# Structure of the ATP-driven methyl-coenzyme M reductase activation complex

Fidel Ramírez-Amador[1,2,9], Sophia Paul[1,2,9], Anuj Kumar[1,2,9], Christian Lorent[3], Sebastian Keller[4], Stefan Bohn[5], Thinh Nguyen[4], Stefano Lometto[6], Dennis Vlegels[6], Jörg Kahnt[6], Darja Deobald[7], Frank Abendroth[2], Olalla Vázquez[1,2], Georg Hochberg[2,6], Silvan Scheller[4], Sven T. Stripp[3,8] & Jan Michael Schuller[1,2 ✉]

Methyl-coenzyme M reductase (MCR) is the enzyme responsible for nearly all biologically generated methane[1]. Its active site comprises coenzyme $F_{430}$, a porphyrin-based cofactor with a central nickel ion that is active exclusively in the Ni(I) state[2,3]. How methanogenic archaea perform the reductive activation of $F_{430}$ represents a major gap in our understanding of one of the most ancient bioenergetic systems in nature. Here we purified and characterized the MCR activation complex from *Methanococcus maripaludis*. McrC, a small subunit encoded in the *mcr* operon, co-purifies with the methanogenic marker proteins Mmp7, Mmp17, Mmp3 and the A2 component. We demonstrated that this complex can activate MCR in vitro in a strictly ATP-dependent manner, enabling the formation of methane. In addition, we determined the cryo-electron microscopy structure of the MCR activation complex exhibiting different functional states with local resolutions reaching 1.8–2.1 Å. Our data revealed three complex iron–sulfur clusters that formed an electron transfer pathway towards $F_{430}$. Topology and electron paramagnetic resonance spectroscopy analyses indicate that these clusters are similar to the [8Fe-9S-C] cluster, a maturation intermediate of the catalytic cofactor in nitrogenase. Altogether, our findings offer insights into the activation mechanism of MCR and prospects on the early evolution of nitrogenase.

Annually, methanogenic archaea release up to 1 Gt of methane ($CH_4$) into the environment[1]. The enzyme responsible for $CH_4$ formation is methyl-coenzyme M reductase (MCR). MCR is one of the most abundant enzymes on Earth and has an important role in the carbon cycle, especially because up to 50% of the $CH_4$ released worldwide is produced by MCR-containing microorganisms, significantly influencing climate change[4]. MCR enzymatically reduces methyl-coenzyme M ($CH_3$–S–CoM) to $CH_4$, with coenzyme B (CoB–SH) as the electron donor for the concomitant formation of CoMS–SCoB heterodisulfide. Similarly, methanotrophs and alkane-oxidizing archaea carry MCR homologues that operate in reverse at considerably lower rates[5,6]. The structure of MCR has been thoroughly described as a 275-kDa heterohexameric complex with an active site that contains coenzyme $F_{430}$, a Ni-coordinated tetrapyrrole cofactor sitting at the interface of subunits McrA, McrB and McrG, arranged in an $A_2B_2G_2$ stoichiometry[2,3]. The nickel ion of $F_{430}$ can exist in the oxidation states of +1, +2 and +3; however, to initiate the catalytic cycle, nickel must be in the Ni(I) state ($MCR_{red1}$). Once isolated, $F_{430}$ is prone to auto-oxidize into the inactive and electron paramagnetic resonance (EPR)-silent Ni(II) state ($MCR_{silent}$), which exhibits remarkable resilience in reverting back to $MCR_{red1}$. The Ni(III) state is referred to as $MCR_{ox1}$ and, similar to $MCR_{red1}$,

shows a characteristic EPR spectrum. By treating cells of *Methanothermobacter marburgensis* with $H_2$, CO or $Na_2S$, it has been possible to reactivate $MCR_{ox1}$ back into the $MCR_{red1}$ state[7–10].

The mechanism by which methanogens achieve the redox potential of the Ni(II)/Ni(I) couple in the $F_{430}$ cofactor ($E^{\circ\prime} = -650$ mV for the isolated cofactor)[11] remains elusive. A clue emerged directly from the *mcrBDCGA* operon. Although the mature complex consists of McrABG, the functions of McrD and McrC were only partially described. McrD has been proposed to facilitate the translocation of $F_{430}$ to the active site of MCR[12]. Moreover, the addition of McrD alleviated product inhibition during the synthesis of $F_{430}$ in vitro, suggesting that it is a possible chaperone[13]. By contrast, McrC has been identified in a multi-protein system, potentially activating $F_{430}$ to the reduced Ni(I) state[14]. Previously, isolated fractions known as A2, A3a and MCR were able to produce $CH_4$ when ATP, vitamin $B_{12}$, Ti(III) citrate, $CH_3$–S–CoM and CoB–SH were supplied in the reaction[15]. Later, the A2 component was established as an ATP-binding cassette (ABC) homologue[16]. A3a comprises several uncharacterized methanogenic marker proteins[17] (Mmp) along with McrC and the multi-enzyme complex Hdr–Mvh–Fwd, suggesting that electron bifurcation may be necessary for the activation process. Nevertheless,

[1]Center for Synthetic Microbiology (SYNMIKRO), Philipps-University Marburg, Marburg, Germany. [2]Department of Chemistry, Philipps-University Marburg, Marburg, Germany. [3]Institute of Chemistry, Technische Universität Berlin, Berlin, Germany. [4]Department of Bioproducts and Biosystems, School of Chemical Engineering, Aalto University, Espoo, Finland. [5]Helmholtz Munich Cryo-Electron Microscopy Platform, Helmholtz Munich, Neuherberg, Germany. [6]Max Planck Institute for Terrestrial Microbiology and Department of Biology, Philipps-University Marburg, Marburg, Germany. [7]Department of Environmental Biotechnology, Helmholtz Centre for Environmental Research (UFZ), Leipzig, Germany. [8]Institute of Chemistry, University of Potsdam, Potsdam, Germany. [9]These authors contributed equally: Fidel Ramírez-Amador, Sophia Paul, Anuj Kumar. ✉e-mail: jan.schuller@synmikro.uni-marburg.de

structural and mechanistic evidence for a protein complex capable of activating $F_{430}$ has yet to be obtained.

Here we report the MCR activation complex of *Methanococcus maripaludis* as an ATP-dependent system. Using redox-controlled cryo-electron microscopy (cryo-EM) single-particle analysis, we resolved a structure (local resolution of 1.8 Å) showing how the A2 component binds to MCR and is potentially involved in rearrangements driven by ATP hydrolysis. We found three complex iron–sulfur (FeS) clusters similar in structure to the L-cluster, a maturation intermediate of the nitrogenase catalytic cofactor[18,19]. These FeS clusters form an unexpected electron transfer pathway to the $F_{430}$ site and may direct low-potential electrons to reduce the nickel ion in $F_{430}$. Overall, our biochemical evidence, cryo-EM structures and spectroscopic data pave the way towards understanding the unique MCR activation machinery.

## ATP drives complex binding and MCR activation

We integrated a Twin-Strep-tag into the genome of the mesophilic, hydrogenotrophic methanogen *M. maripaludis*[20] at the N terminus of McrC (Supplementary Tables 1 and 2 and Extended Data Fig. 1), and the modified strain showed no growth perturbations when compared to the wild type. By affinity purification of Twin-Strep McrC (Fig. 1a), the MCR core subunits McrA (61 kDa), McrB (46.7 kDa) and McrG (29.6 kDa) were pulled down. Along with Twin-Strep McrC (24.3 kDa), the methanogenic marker proteins Mmp7 (34.9 kDa), Mmp17 (21.1 kDa), Mmp3 (56.5 kDa), A2 component (59.5 kDa) and a protein of unknown function (DUF2098; 10.6 kDa) were also co-purified. Twin-Strep McrC was verified by western blot using an anti-Twin-Strep-tag antibody (Fig. 1a) and mass spectrometry, confirming all the co-purified subunits and post-translational modifications in McrA (Supplementary Table 3 and Extended Data Fig. 2a). The complex migrated as two major peaks (280 nm) during size exclusion chromatography (SEC; Fig. 1b). Both peaks showed absorbance at 430 nm owing to the presence of $F_{430}$, but only the larger peak (approximately 500 kDa) had a signal at 330 nm, indicating potential FeS clusters[21] in the MCR activation complex. The smaller SEC peak (approximately 298 kDa) is in good agreement with that of the MCR core alone (Fig. 1b and Extended Data Fig. 2b,c). Ultraviolet–visible spectroscopy (UV–Vis) spectra showed that both the activation complex-bound MCR and the MCR core shared a maximum absorbance at around 420 nm and a shoulder at 447 nm, which was more pronounced in the MCR core (Extended Data Fig. 2d,e). These electronic features are consistent with those of Ni(II)–$MCR_{silent}$[8,9,22], leading us to conclude that most of the $F_{430}$ rests in the Ni(II) state in both the biogenesis intermediate and MCR. However, the Ni(III)–$MCR_{ox1}$ features in the UV–Vis spectrum cannot be discarded either (see below).

To investigate the ability of the purified complex to activate MCR in an ATP-dependent manner, we established an in vitro assay following the production of $CH_4$ (Fig. 1c). Before the reaction, 5 μM of the enzyme was preactivated at 37 °C and pH 7 in the presence of Ti(III) citrate, $CH_3$–S–CoM and ATP to avoid inhibition by the immediate formation of CoMS–SCoB[14,23] (Methods). Then, 12 μg of activated protein was added to the mixture containing $CH_3$–S–CoM, CoB–SH and the remaining reaction components. The reaction resulted in 40 ± 6 nmol of $CH_4$ after 2 h at 37 °C and pH 7 when the sample was preincubated in the presence of 5 mM ATP, doubling the $CH_4$ produced by the non-activated enzyme (Extended Data Fig. 2f). Without external addition of ATP ('as isolated'), only 9 ± 2 nmol of $CH_4$ was produced (Fig. 1c). Strikingly, when alkaline phosphatase was included to deplete the ATP in the reaction, no $CH_4$ production was detected. Hence, under the assay conditions, Ti(III) citrate at pH 7 cannot reduce the nickel ion in $F_{430}$ unless ATP is present. Thus, the basal production of $CH_4$ is probably due to trace amounts of ATP co-purified with the protein complex. Although further attempts to determine activity rates were hindered because of the low protein yields (less than 0.5 mg of the activation complex

from 10 l of culture), our activity assays proved that the generation of $CH_4$ is clearly ATP-dependent, requiring a minimal set of necessary components for the activation of MCR.

We analysed the effect of ATP on the formation of the activation complex using mass photometry, a single-molecule technique (Extended Data Fig. 2g). For the 'as isolated' sample, mass photometry confirmed an assembly of 502 kDa and defined peaks of smaller size, indicating partial disruption when extensively diluted (0.03 μM protein). ATP complementation stabilized the binding of the smaller subunits to the larger MCR activation complex. A similar effect was observed when AMP-PNP, a non-hydrolysable ATP analogue, was used instead of ATP. This demonstrates that ATP binding promotes the assembly of the A2 component with MCR, reinforcing the crucial function of A2 for full complex formation.

## Architecture of the MCR activation complex

To reveal the molecular structure of the MCR activation complex, we vitrified the protein 'as isolated' and after incubating with 5 mM ATP under redox-controlled anaerobic conditions (Methods), before performing cryo-EM single-particle analysis. Addressing preferred orientation effects, we used a combination of non-tilted and 20° pre-tilted micrographs for the 'as isolated' sample (Extended Data Fig. 3), whereas only 25° pre-tilted micrographs were used for the complex incubated with ATP (Extended Data Fig. 4). Our analysis revealed two different assembly intermediates from the 'as isolated' dataset. One map harbours the full activation complex, including the A2 component (58% of particles) at a resolution of 2.5 Å, whereas the second map represents an assembly lacking the A2 subunit (42% of particles), refined to a resolution of 2.7 Å. In line with our mass photometry observations, the occupancy of the A2 component in the sample preincubated with ATP drastically increased (92% of the particles). This dataset allowed us to refine a structure with a global resolution of 2.1 Å (Fig. 1d). Furthermore, after masked 3D classification and local refinement steps, the protein region containing $F_{430}$ and FeS clusters reached a local resolution of 1.8 Å (Extended Data Fig. 5). Therefore, the previously observed role of ATP in the stabilization of the A2 component was further confirmed.

Our structures revealed that the activation complex binds asymmetrically to only one of the MCR heterotrimers (Fig. 1d and Extended Data Fig. 6a). The Mmp proteins collectively encircled one side of the MCR core, with the heart of this complex formed by McrC, Mmp7, Mmp17 (partially resolved; Extended Data Fig. 6b) and Mmp3, which held the core subunits together. McrC used a flexible loop (C138 to I168) to bind DUF2098 (Extended Data Fig. 6c), whereas a second loop ($V72^{McrC}$ to $K95^{McrC}$) wedged into the N-terminal helix of the more proximal McrA, which became disordered and created a groove towards the $F_{430}$ at the active site (Fig. 2a,b). This unfolded N-terminal helix of McrA was held in place by van der Waals contacts from $Y404/F405^{Mmp3}$ together with $P396/F399^{A2}$, mainly sandwiching $L12/F16^{McrA}$ (Fig. 2c). In *Methanosarcina acetivorans*, the same region of McrA is recognized by McrD before the translocation of $F_{430}$ (ref. 12). Therefore, this latch within McrA emerged as a key interaction site with the activation machinery, regulating access to the $F_{430}$ binding cavity.

The A2 component consisted of two ABC family domains arranged antiparallel to each other (Extended Data Fig. 6d). A2 sits on the backside of MCR binding McrG through electrostatic interactions, without contacting the other activation subunits (Fig. 2d and Extended Data Fig. 6e). The N-terminal region of A2 contains a highly flexible loop coordinating a Zn ion, which is conserved among methanogenic and methanotrophic species (Extended Data Fig. 6f). Our structure revealed two ATP molecules and their coordinating Mg ions (Fig. 2d), indicating that the structure captured the enzyme in a pre-hydrolytic state. This finding reinforces the basal production of $CH_4$ when ATP was not added in vitro (Fig. 1c).

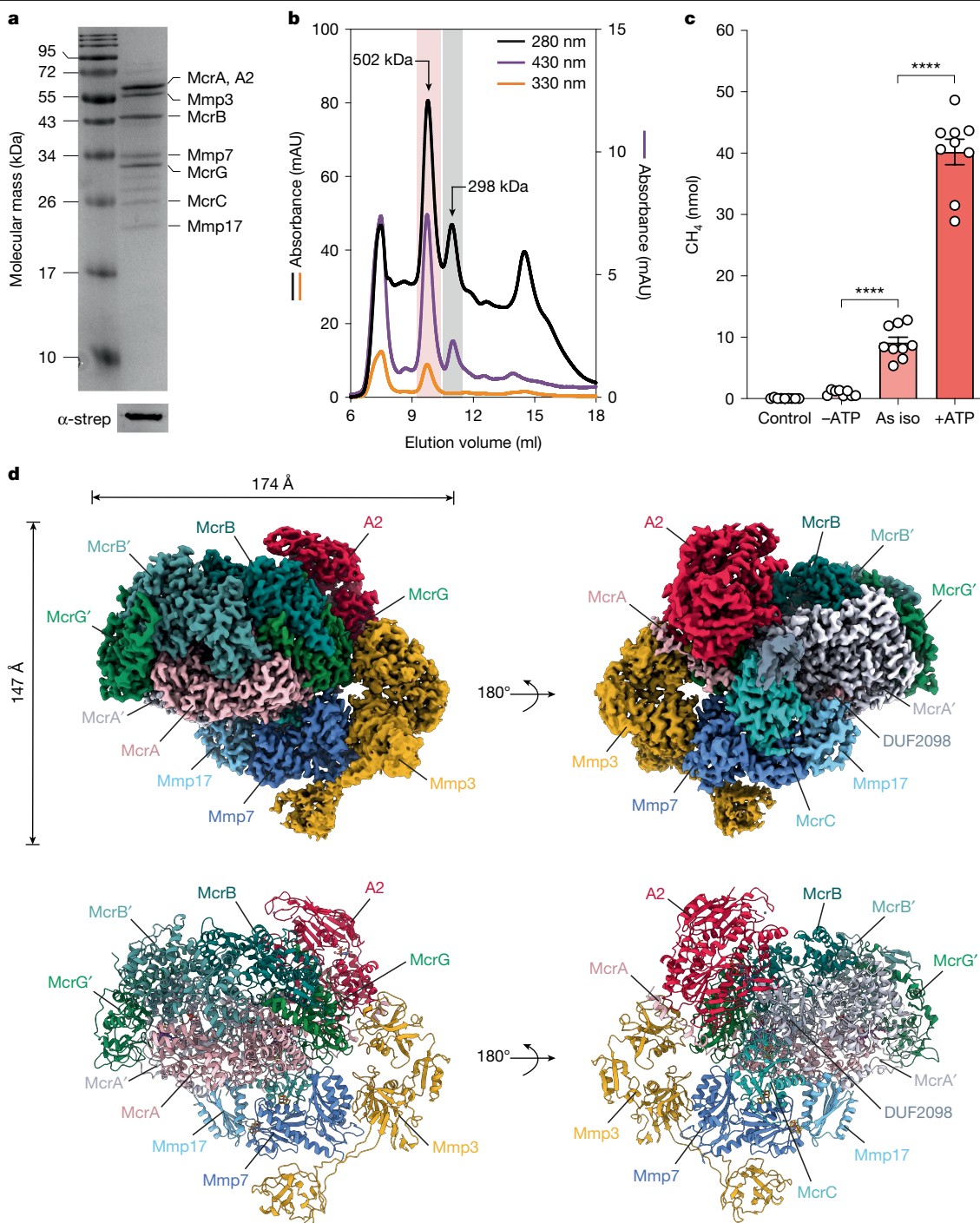

**Fig. 1 | Composition, ATP-dependent in vitro activation and molecular architecture of the MCR activation complex. a**, Purified subunits of the MCR activation complex and western blotting of Twin-Strep McrC using an anti-Twin-Strep-tag antibody. At least three independent polyacrylamide gels exhibited the same separation pattern (for uncropped gels, see Supplementary Fig. 1). **b**, Separation of the MCR activation complex and MCR core by SEC (Extended Data Fig. 2b,c). **c**, CH$_4$ production assays showing the ATP-dependent activation of MCR in vitro. Control, reaction with no protein; −ATP, depletion of ATP with

alkaline phosphatase (Quick CIP; Methods); As iso, 'as isolated' protein; +ATP, external addition of ATP. All data points are presented as mean values ± s.e.m. (error bars) from three independent biological replicates, each with three technical replicates. Statistical analysis was performed using an unpaired *t*-test to compare the reaction without ATP (−ATP) versus 'As iso' (significance level $P \leq 0.0001$) and +ATP versus 'As iso' (significance level $P \leq 0.0001$). **d**, Segmented cryo-EM map (top) and corresponding protein model (bottom) of the MCR activation complex.

## Functional asymmetry in MCR during activation

Our model unveils a functional asymmetry within both active sites of the MCR core (Fig. 3 and Extended Data Fig. 7). Both heterotrimers contained an F$_{430}$ molecule, fitting neatly into a pocket formed by each McrABG protomer. We defined the activation complex binding to the

'proximal side' of MCR. At the distal side ('), the architecture was consistent with the reported structures of MCR[2,3,12] (Fig. 3a). Axial coordination of the Ni ion in F$_{430}$ by both Q151$^{McrA}$ and the thiol in CoM−SH was present, with the latter being stabilized by Y336$^{McrA'}$ and Y367$^{McrB'}$ hydroxy groups. Similarly, the phosphate moiety of CoB−SH was kept in place by the positively charged McrA residues R229, K260 and methylated H261.

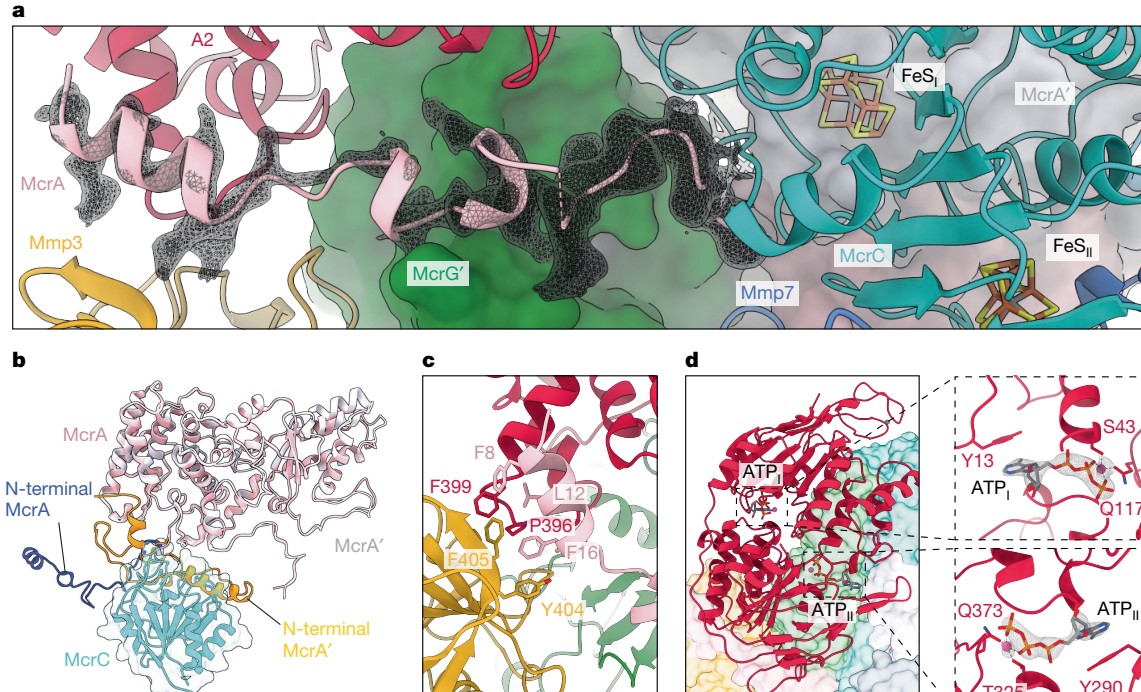

**Fig. 2 | Interactions between MCR and the activation complex proteins.**
**a**, Bottom view of the disordered N-terminal helix of McrA relative to McrC and the activation complex. Owing to the high flexibility, residues 31–58 of McrA are missing in the electron density (dashed line). The MCR core is depicted as a surface; only the unfolded region of McrA and the activation subunits are shown as cartoons. The electron density is depicted as a mesh (contour level $\sigma$ of 0.04). **b**, Overlay of McrA and McrA′ (proximal and distal to the activation complex, respectively) showing the displacement and loss of secondary structure from the N-terminal section of McrA. **c**, Hydrophobic patches formed by the C-terminal domain of Mmp3 and the A2 component to hold the disordered N-terminal helix of McrA. **d**, A2 component (cartoon) sitting on the backside of MCR (surface). ATP molecules bind within both active sites, suggesting a pre-hydrolytic state (contour level $\sigma$ of 0.08). Mg ions are shown as pink spheres.

In the proximal active site, we found the CoMS–SCoB heterodisulfide next to $F_{430}$ (Fig. 3b). However, the product interacted with the Ni ion in a significantly different manner than earlier crystal structures[3,24] (Extended Data Fig. 7a), in which the sulfonate oxygen atoms of CoM–SH are perpendicularly oriented towards the tetrapyrrole plane of $F_{430}$ within a narrow hydrophobic pocket. In our model, the position of the CoM–SH fraction runs parallel to the tetrapyrrole plane, nearly mirroring the distal side. In the middle, the newly formed S–S bond is in close distance to the Ni ion (Fig. 3b). This coordination is particularly relevant, representing a nascent disulfide intermediate discussed in all proposed MCR mechanisms[8,25]. Around the heptanoyl chain of CoMS–SCoB, displacements above 3 Å (more than 10 Å in the 'as isolated' structure) of loops F361-G371$^{McrB}$ and K244-E249$^{McrA′}$ enabled the CoB moiety to freely descend through the active pocket (Fig. 3c–e and Extended Data Fig. 7b). Such conformational changes directly impact the CoB fraction, which becomes very flexible as implied by the electron density maps (Extended Data Fig. 5b). On the opposite side of $F_{430}$, a 5.5-Å rearrangement in the loop containing the Ni-coordinating Q151$^{McrA′}$ avoids its canonical axial coordination (Fig. 3b,e). Hydrophobic interactions between L88/I79$^{McrC}$ and V150$^{McrA′}$ facilitated the formation of a groove at the N-terminal region of McrA (Fig. 2a and Extended Data Fig. 7c,d). Additionally, the electrostatic contacts of R177$^{McrC}$ with E156$^{McrA′}$ stabilized the opening. This remodelling may permit both the reduction of $F_{430}$ and further release of CoMS–SCoB from the active site.

When compared to the structure without the A2 component (Extended Data Fig. 6a), no significant differences were found in the active site or wedging loop of McrC. However, in the absence of A2 (Fig. 2c), the N-terminal helix of McrA became completely diffusive (Extended Data Fig. 6a). Altogether, this unique ATPase may not only stabilize the opening latch of McrA but also facilitates ATP-dependent collective motions in the activation complex subunits (Fig. 3 and Extended Data Figs. 3c and 5a).

## Nitrogenase-like FeS clusters pathway

Shaping a potential electron transfer pathway towards $F_{430}$, we observed three electron-rich regions at the interface of McrC, Mmp7 and Mmp17 (Fig. 4a–c and Extended Data Fig. 8a). These densities neatly accommodated two [4Fe-3S] clusters bridged by three non-protein ligands, which formed a belt perpendicular to the axis defined by the two coordinating residues (Extended Data Fig. 8b). The overall arrangement and composition closely resembled the catalytic cofactor of nitrogenase—the M-cluster[26,27] (Extended Data Fig. 8c). The three 'belt sulfur' atoms of the M-cluster[28] perfectly fit the density features bridging the [4Fe-3S] fragments. In addition, a weaker electron density at the centre of the M-cluster suggests a coordinating carbide ion[29]. Similar features have been observed in all cryo-EM structures of nitrogenases[30–32] (Extended Data Fig. 8d). Although the local resolution of these cofactors reached 1.8 Å (Extended Data Fig. 5a), further experimental validation is required for definitive assignment of the carbide ion. An average of $27 \pm 2$ Fe ions, identified by inductively coupled plasma–mass spectrometry (Supplementary Table 4), supported the presence of three [8Fe-9S-C] clusters, whereas the absence of stoichiometric amounts of Mo or V, besides the lack of the homocitrate ligand, excluded a typical nitrogenase M-cluster[26,27,33]. From our high-resolution map, we measured the atom-to-atom distances of Fe–Fe (average of 2.61 Å), Fe–S (average of 2.25 Å) and Fe–C (average of 1.99 Å) (Fig. 4c). These values are fairly similar to those of the M-cluster (Supplementary Table 5) but contrast with another 'bridged' cluster, such as the K-cluster (early intermediate of M-cluster)[34]. Overall, the topology and composition highly matched those of the L-cluster[18,19], the [8Fe-9S-C] precursor of the M-cluster in the biosynthesis of nitrogenase.

The protein environment around these FeS clusters, from now on referred to as FeS$_I$, FeS$_{II}$ and FeS$_{III}$ (Fig. 4a,b), is primarily provided by

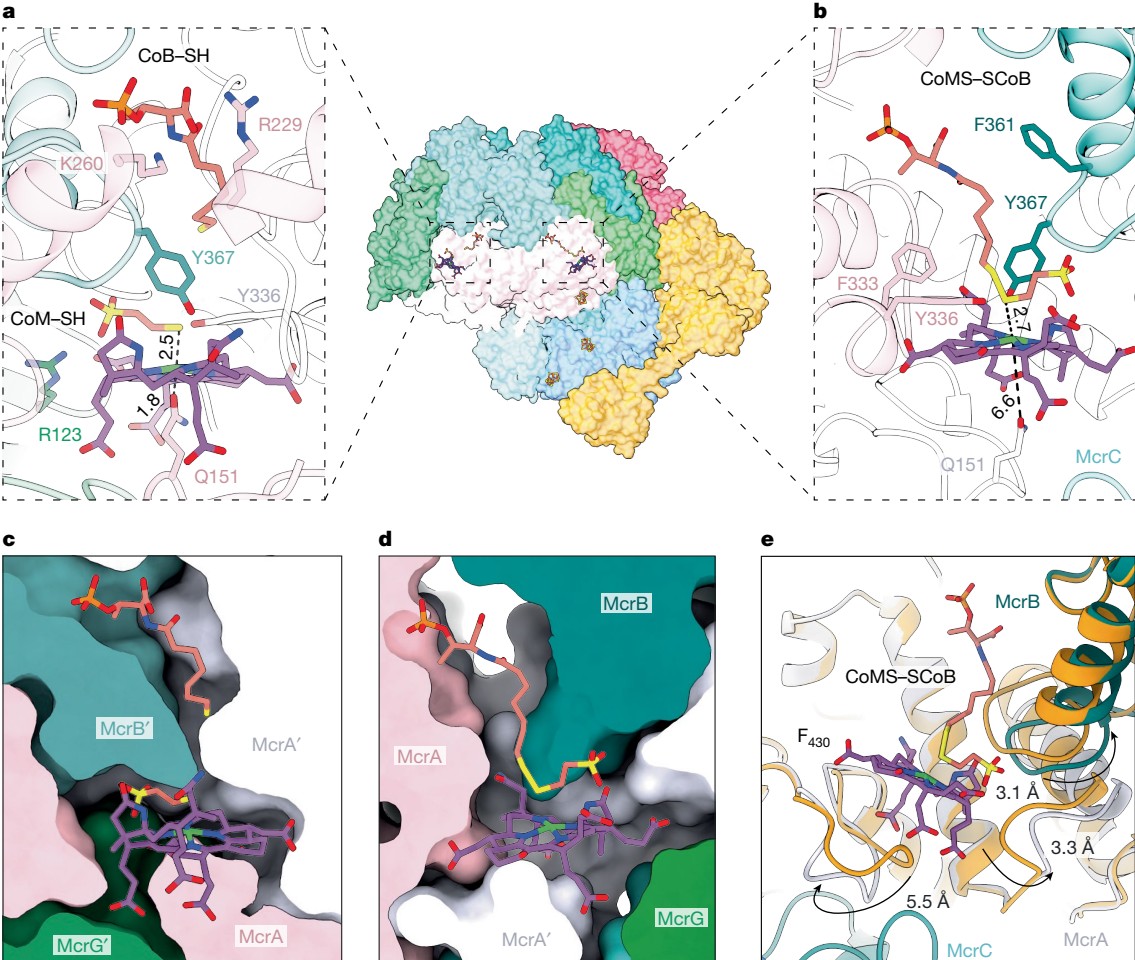

**Fig. 3 | Asymmetric catalytic sites during the activation of MCR. a**, Distal active site showing the substrates CoB–SH and CoM–SH. **b**, Proximal active site containing the CoMS–SCoB by-product. Key residues rearranged to expand the pocket surrounding CoMS–SCoB. All distances between relevant atoms (dashed lines) are in angstrom (Å). **c**,**d**, Comparison of the structural arrangements at the distal (′) (**c**) and proximal (**d**) catalytic sites of MCR. MCR is depicted as a surface. **e**, Superposition of distal (orange) and proximal MCR subunits relative to the activation complex. The indicated loops in McrA and McrB are reoriented when McrC and the activation complex bind.

McrC and the McrC-like domain of Mmp7. Similarities between McrC and Mmp7 became apparent when superimposed (Extended Data Fig. 8e), despite their sequence identity (26.3%) and size difference (Supplementary Table 3). The three clusters were nested in grooves reminiscent of the secondary coordination sphere surrounding the M-cluster in nitrogenases[35], mostly because of the abundant hydrophobic and hydrogen-bonding residues (Extended Data Fig. 8e,f). However, they show specific differences in coordination. $FeS_I$ is located in McrC, about 9 Å away from the closest carboxylate group in $F_{430}$ (Fig. 4a,b) and in good distance for rapid electron transfer[36]. Strikingly, both terminal Fe ions of $FeS_I$ are ligated to histidines, namely H49[McrC] (Fe1) and H118[McrC] (Fe8), which would substantially reduce the redox potential compared to cysteine coordination[37]. In nitrogenases, the M-cluster is covalently bound to a cysteine and coordinated by a histidine[30–32]. $FeS_{II}$ has unique coordination (Fig. 4b) with both residues being cysteines, namely C55[McrC] (Fe1) and C12[McrC] (Fe8). Owing to its close proximity, C90[Mmp7] could additionally serve as a bridging ligand between Fe3 and Fe7 in $FeS_{II}$ (Extended Data Fig. 8g). However, this would demand replacement of the belt sulfur atom that is also present in $FeS_I$ and $FeS_{III}$. Therefore, we cannot rule out C90[Mmp7] coordinating $FeS_{II}$. Most distal from the active site, the $FeS_{III}$ cluster was near the surface (Fig. 4a,b), specifically at the interface of Mmp7 and Mmp17. In this case, Fe1 is coordinated by C143[Mmp7], whereas H84[Mmp7] ligates to Fe8 in a topology closely resembling that of the M-cluster in nitrogenases[30–32]. Overall, we

assume that differences in coordination modulate the redox properties of these FeS clusters and allow for directional electron transfer.

To explore the electronic nature of the cofactors in the MCR activation complex, we used EPR spectroscopy (Fig. 4d and Extended Data Fig. 8h–k). Two different EPR signals can be distinguished, exhibiting an unambiguous difference in their temperature dependence. The low-field signal ($g_1 = 2.23$, $g_2 = 2.17$ and $g_3 = 2.15$) was most prominent at higher temperatures and can be assigned to the $MCR_{ox1}$ state, accepted to form a Ni(III) thiolate potentially in equilibrium with a high-spin Ni(II) coupled to a thiyl radical form[38,39]. The results of previous studies indicate that the EPR signal of $MCR_{ox1}$ may remain strong even under reducing conditions and may coexist with Ni(I) or Ni(II) species[8,9]. Thus, on the basis of the EPR data, we observed minor amounts of $MCR_{ox1}$ in our sample. However, as evidenced by the UV–Vis bands (Extended Data Fig. 2d,e), a major part of the $F_{430}$ in our sample seemed to rest in an EPR-inactive Ni(II)–$MCR_{silent}$ state. In the high-field part of the EPR spectrum (between $g = 2.15$ and 1.70), a complex signal appeared at low temperatures. This signal probably originated from a multitude of paramagnetic FeS cluster species that resemble the maturation intermediates of the M-cluster[26,40]. Differences in the spectral pattern, compared to typical K- and L-cluster-type species[26], are probably related to variations in both coordination and structure, for example, the two-fold histidine ligation of $FeS_I$ (Fig. 4b). Because there were no indications of additional high-spin states in the low-field region (Extended Data

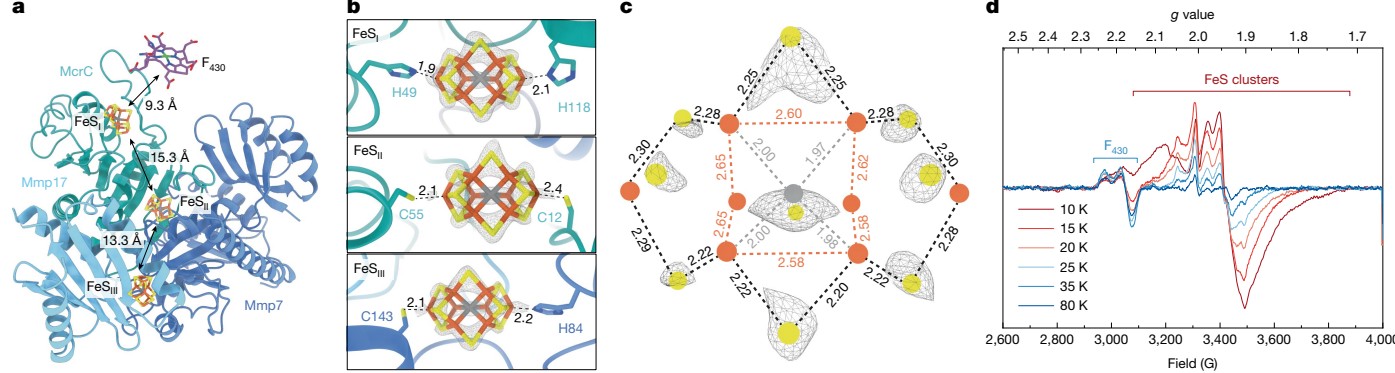

**Fig. 4 | Nitrogenase-like FeS clusters are coordinated by McrC and Mmp7.** **a**, Edge-to-edge distances between the FeS clusters and $F_{430}$ at the proximal active site. **b**, FeS clusters in their electron density and coordinating residues (contour levels $\sigma$ of 0.168 for $FeS_I$ and 0.235 for $FeS_{II}$ and $FeS_{III}$). **c**, Electron density cloud of sulfur atoms in $FeS_{II}$ at a contour level $\sigma$ of 0.32. Selected bond

lengths are shown as dashed lines, and colour depends on the atom pair (Fe–Fe in orange, Fe–S in black and Fe–C in grey). In **b** and **c**, distances are in angstrom (Å). **d**, Temperature-dependent EPR spectra of the 'as isolated' MCR activation complex recorded with 1-mW microwave power.

Fig. 8h), all signals seemed to arise from $S = \frac{1}{2}$ ground state, which is also characteristic of K- and L-clusters[27]. Notably, the FeS clusters were interacting magnetically, which can be derived from the significant broadening of the signals when lowering the temperature and power-dependent saturation curve (Extended Data Fig. 8i). The presence of the reductant sodium dithionite had a minor effect on the FeS cluster signals but clearly lowered the intensity of the Ni(III)–MCR$_{ox1}$ signal (Extended Data Fig. 8j). By contrast, only weakly paramagnetic FeS species were present after incubating the protein with the oxidant thionine (Extended Data Fig. 8k), indicating that most of the clusters were diamagnetic under mildly oxidizing conditions. Although further investigation is required to unveil how these FeS clusters specifically operate, their arrangement and coordination imply a role in electron transfer during the activation of MCR.

## MCR activation and nitrogenase evolution

The presence of [8Fe-9S-C] clusters had only been described in proteins belonging to the Nif family, where they compose the catalytic cofactor of nitrogenases and nitrogenase-like reductases[30–32,41]. Our discovery of potential [8Fe-9S-C] clusters in the MCR activation complex raised the following question: in which of these enzymatic systems did such clusters come first, MCR or the Nif family? To assess their phylogenetic origin, we reconstructed phylogenetic trees for McrC and Mmp7, which carry [8Fe-9S-C] cluster-binding sites (Extended Data Fig. 9a,b). Both proteins are present in Euryarchaeota; members of the TACK group (named after Thaumarchaeota, Aigarchaeota, Crenarchaeota and Korarchaeota); and Asgard Archaea, with individual tree topologies that each broadly align with established archaeal phylogenies[42,43]. This supports their presence in the last common ancestor of these archaeal lineages, consistent with other studies that also found MCR to be present in this ancestor as well[44]. We also reconstructed a phylogeny of NifB (Extended Data Fig. 9c), the enzyme that synthesizes [8Fe-9S-C] clusters used by the Nif family[45]. NifB has been shown to be essential in other methanogens, even those lacking nitrogenase-encoding genes[46–48], and is probably involved in the formation of FeS clusters for the activation of MCR. Rooting this tree proved challenging because of several duplications and horizontal gene transfers. Nevertheless, NifB is also present in Euryarchaeota, TACK and Asgard archaea, suggesting that it originated at least as early as MCR, McrC and Mmp7. We next investigated whether the Nif family could be as old as MCR and its activation machinery. In addition to nitrogenases and nitrogenase-like proteins, the Nif family includes CfbD, a very distant homologue that takes part in the biosynthesis of coenzyme $F_{430}$ and does not contain [8Fe-9S-C]

clusters[13,49]. The phylogeny of CfbD resembled that of McrC and Mmp7 (Extended Data Fig. 9d), with an origin in the last common ancestor of TACK, Asgard and Euryarchaeota. CfbD is probably the oldest branch on the Nif phylogeny because all other Nif homologues resulted from a series of gene and operon duplications that the homodimeric CfbD did not undergo (Extended Data Fig. 9e). Furthermore, we inferred a separate phylogeny for all members of the Nif family that are either suggested to contain [8Fe-9S-C] clusters or to act on their maturation process (Extended Data Fig. 9f). Our phylogenetic analysis implies a root of the wide Nif family in Euryarchaeota because we found only very scattered evidence of TACK or Asgard archaeal members, each of which was a clear horizontal transfer from Euryarchaeota. No evidence was found for [8Fe-9S-C] cluster-containing Nif proteins in the last common ancestor of TACK, Asgard and Euryarchaeota, pointing to an origin either at the base or inside Euryarchaeota. Together, our phylogenetic results imply that the [8Fe-9S-C] cluster was first used for MCR activation and later incorporated into nitrogenases and nitrogenase-like systems (Extended Data Fig. 9g).

## Discussion

We validated that our purified complex activates MCR through an ATP-dependent process, and the cryo-EM structure confirmed the involvement of the A2 component (an ABC-type ATPase)[16]. Upon the addition of Ti(III) citrate, previous studies on the activation of MCR[15,23] achieved a maximum activity of 0.1 µmol $CH_4$ min$^{-1}$ mg$^{-1}$ (or 0.1 U mg$^{-1}$), not much different from our results after 2 h of reaction (0.028 U mg$^{-1}$). Nevertheless, such values remain low compared to the average activity of mature MCR from *Methanothermobacter* species (20 U mg$^{-1}$ (ref. 22), with 100 U mg$^{-1}$ being the maximum ever reported[8]). By contrast, our purification strategy captured a low-abundance and highly transient biogenesis intermediate, challenging a direct comparison with a fully mature MCR. This discrepancy indicates an incomplete understanding of the activation process, in which additional factors might be required to efficiently reduce the nickel in $F_{430}$. Accordingly, no activity was detected when Ti(III) citrate was replaced by dithiothreitol (DTT) (Extended Data Fig. 2f), as reported previously[14], suggesting that auxiliary proteins in the A3a fraction may be critical for lowering the redox potential barrier to activate $F_{430}$. In *M. marburgensis*, genes encoding for more Mmp proteins are clustered in a single open reading frame (MTBMA c04360–c04410, followed by *nifB*), whereas *Methanosarcina* species even contain the A2 component gene within an equivalent locus. Therefore, homologues of these extra Mmp proteins are presumably required in

*M. maripaludis* for the physiological activation of $F_{430}$, although they are widely dispersed around the genome.

We suggest that the role of the A2 component extends beyond stabilizing the activation complex, making $F_{430}$ accessible within the active site (Fig. 2). From this perspective, the binding of ATP in A2 facilitates interactions with MCR and the activation of $F_{430}$, as confirmed by the production of $CH_4$. When ATP was replaced by AMP-PNP, not only the stabilization persisted but a moderate increase in the $CH_4$ concentration was observed (Extended Data Fig. 2f,g). This suggests that conformational states influenced by nucleotide binding (like the lost coordination of $Q151^{McRA'}$) already affected activation and potentially modulated the redox potential of $F_{430}$. As a free ligand, $F_{430}$ has a midpoint redox potential of $-650$ mV (ref. 11), and Ti(III) citrate at pH 9–10 had shown enough reducing power for its activation[8,9]. However, the binding and hydrolysis of ATP could theoretically increase the redox potential, such that Ti(III) citrate ($E^{o'} = -480$ mV at pH 7) (ref. 50) reduced Ni(II) to Ni(I). Comparably, during the ATP-driven activation of some corrinoids, ATP hydrolysis drastically increased the midpoint redox potential of the Co(II)/Co(I) couple from $-500$ to $-250$ mV (ref. 51). Moreover, more Mmp interactors may push the redox potential of $F_{430}$ even further, allowing the reduction of nickel by milder reducing agents, such as DTT ($E^{o'} = -320$ mV) (ref. 14), and physiological electron donors, such as ferredoxin ($E^{o'} = $ approximately $-400$ mV) (ref. 52). Altogether, this places the function of the A2 component alongside other prominent nucleotide-switching machines that can drive either catalytic activation or product release[53,54]. Although further details on the redox cycle of $F_{430}$ are still unknown, ATP binding and hydrolysis seem to trigger such a complex mechanism.

In addition to the activation process, our findings offer insights into the catalytic mechanism of MCR. So far, no other structure has captured different enzymatic states within the same MCR core (Fig. 3). This observation aligns with the 'two-stroke engine' mechanism[55], proposing that conformational changes induced by one active site directly influence the other inactive site. In our model, this allostery is reflected by an expanded proximal active site holding CoMS–SCoB, in contrast to the rather narrow distal pocket containing the substrates. This proximal site reorganization agrees with previous $^{19}$F-electron–nuclear double resonance spectroscopy[56] results, in which a less rigid structural framework was necessary for CoB–SH to approach the Ni ion in $F_{430}$. Furthermore, the proposed mechanism of MCR theorizes a heterodisulfide anion radical (CoMS–SCoB$^{\cdot}$) that reduces Ni(II) back to Ni(I), followed by product release[25]. Although we cannot directly observe a radical intermediate, our structure revealed a CoMS–SCoB inhibited state[14,23], which partially explains the Ni(III)–$MCR_{ox1}$ signal observed by EPR[57] (Fig. 4d). Additionally, product release might be hindered by the binding of the activation machinery. In general, our findings shed light on a previously unknown link between the catalytic mechanism and activation process of MCR.

Strikingly, three uniquely coordinated FeS clusters (Fig. 4) constitute an electron transfer pathway towards $F_{430}$. Whereas the high-resolution electron density of such cofactors clearly suggests the [8Fe-9S-C] topology of an L-cluster (Fig. 4 and Supplementary Table 5), the electron weak interstitial carbide ion requires thorough spectroscopic efforts for validation. Techniques, such as valence-to-core X-ray emission spectroscopy[19], electron spin echo envelope modulation[29] and $C^{13}$-labelling coupled to electron–nuclear double resonance spectroscopy[58], have successfully identified the carbide signal in both M- and L-clusters. Similar methods can be used to characterize the metallocofactors binding the MCR activation complex. Moreover, our structure shows the existence of these clusters beyond the Nif family members, providing a critical missing link in the evolution of nitrogenases. We propose that the [8Fe-9S-C] cluster originated first in the methanogenesis context, specifically for the activation of MCR. As suggested by our phylogenetic reconstructions (Extended Data Fig. 9), these cofactors were later incorporated into a protein scaffold most likely derived from CfbD, whose relationship to nitrogenases has been previously known[59,60]. Thus, the protein scaffold and catalytic cofactors of the nitrogenase family (like FeMoco) would ultimately derive from the same physiological function: the biosynthesis and activation of coenzyme $F_{430}$.

Overall, our high-resolution structure of the strictly ATP-dependent activation complex of MCR provides a platform for investigating this process in similar systems, that is, methanotrophic or alkane-oxidizing archaea. Understanding such mechanistic details is crucial for more robust strategies about bioengineering of $CH_4$-converting organisms or the heterologous production of MCR, ultimately aiming to mitigate $CH_4$ emissions.

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

## Methods

### Plasmids and *M. maripaludis* transformation

We performed genome editing in *M. maripaludis* JJ Δupt (referred to as *M. maripaludis*) using the CRISPR–*Lb*Cas12a system[20]. In brief, suitable genome-targeting guide RNA (gRNA) sequences were designed in CHOPCHOP[61] (https://chopchop.cbu.uib.no/), setting the parameters for Cas12a (Cpf1) and TTTN as the 5′PAM. The DNA sequence of *mcrC* (MMJJ_12790) and the reference genome of *M. maripaludis* JJ (DSM 2067) were used as inputs. From this prediction, oligonucleotide fragments were designed, including 5′-*Paq*CI digestion sites, followed by the gRNA lacking the protospacer adjacent motif (PAM) sequence (Supplementary Table 1). Both primers were annealed at 94 °C for 2 min and cloned into the shuttle plasmid pMM002p[20] (Supplementary Table 2) through Golden Gate Assembly. A synthetic fragment containing the coding sequence of a Twin-Strep-tag in the 5′ sequence of *mcrC*, including 750-bp homology extensions upstream and downstream, was designed and further synthesized (Twist Bioscience). The synthetic block was then cloned through Gibson Assembly into the pMM002p/gRNA plasmid and transformed in *Escherichia coli* DH5α competent cells, resulting in the plasmid pMM002p/TS-*mcrC*. All cloning steps were confirmed by sequencing (Microsynth AG). For the transformation of *M. maripaludis*, 2 µg of plasmid was added to 5 ml of cells (OD$_{600}$ of approximately 0.8) in McC medium[62] and immediately overpressurized to 1.5 bar of H$_2$/CO$_2$/H$_2$S (75:20:5) before incubating for 4 h at 37 °C and 200 rpm. The recovered cells were centrifuged at 1,500$g$ for 2 min at room temperature, resuspended in 100 µl of fresh McC medium and plated on solid McC with 2.5 µg ml$^{-1}$ of puromycin. The plates were stored in a tightly closed anaerobic chamber overpressurized to 1.5 bar of the same gas mix as before. Colonies were obtained after 3–5 days of incubation at 37 °C and further confirmed by colony polymerase chain reaction (PCR). To curate *M. maripaludis* from the pMM002p/TS-McrC plasmid, selected colonies were inoculated and grown in McF medium[62] free of antibiotics for 3 days and then plated on solid McC containing 0.25 mg ml$^{-1}$ of 6-azauracil. Finally, the curated colonies were confirmed by colony PCR and sequencing of the Twin-Strep-containing region (Extended Data Fig. 1).

### Culture conditions, expression and purification of the MCR activation complex

Cultures of *M. maripaludis* expressing Twin-Strep McrC were grown at 37 °C and 180 rpm in 10 l of McC media until they reached an OD$_{600}$ of 0.8 (late exponential phase; Extended Data Fig. 1d). Initially, pre-cultures in serum bottles with 100 ml of McC were used to inoculate 700 ml of production medium in 2-l bottles sealed with rubber stoppers. Before adding cells, all cultures were reduced with 2.5% Na$_2$S (1 ml l$^{-1}$) and further inoculated with 2% of the cell volume. The bottle headspace was filled with 1.5 bar of H$_2$/CO$_2$ (80:20) and flushed every 12 h until collection. All further steps were performed under strict anaerobic conditions in a vinyl anaerobic chamber (Coy Laboratory Products) filled with 95% N$_2$ and 5% H$_2$. The cells were collected by centrifugation at 12,000$g$ for 35 min at 4 °C before being resuspended in buffer A (20 mM HEPES and 150 mM NaCl; pH 7.6) with 1 mM phenylmethylsulfonyl fluoride (PMSF). The cells were then lysed by sonication following 30 s ON/30 s OFF cycles for 10 min on ice with output and maximum power of 5 and 60%, respectively. The lysate was centrifuged at 17,000$g$ for 35 min at 4 °C and further purified using a gravity column containing Strep-Tactin Sepharose resin (IBA Lifesciences). Finally, the eluted fractions were recovered with buffer A plus 2.5 mM desthiobiotin and concentrated using a 30-kDa Amicon Ultra centrifugal filter (Merck Millipore). Protein concentration was determined by both Bradford and bicinchoninic acid assays, on the basis of a calibration curve using bovine serum albumin. We used 15% sodium dodecyl sulfate (SDS) gels to assess protein quality, and we performed a semi-dry western blot with a horseradish peroxidase (HRP)-coupled anti-Twin-Strep-tag antibody (1.5 mg ml$^{-1}$; IBA Lifesciences) to visualize and confirm the presence of Twin-Strep McrC.

### MCR activation and activity assay

The activation of MCR was assayed following the production of CH$_4$. In brief, 5 µM enzyme was preincubated with buffer B (20 mM HEPES, 150 mM NaCl and 15 mM MgCl$_2$; pH 7) supplemented with 10 mM CH$_3$–S–CoM, 20 mM Ti(III) citrate and 5 mM ATP (optional) for 30 min at 37 °C before placing the tubes on ice. Other conditions, such as 5 mM AMP-PNP, 2.5 U Quick CIP (New England Biolabs) or 5 mM DTT (instead of Ti(III) citrate), were also tested under the same pre-incubation conditions. For the activity assay, 12 µg of preincubated enzyme was mixed in fresh buffer B plus 10 mM CH$_3$–S–CoM, 2 mM CoB–SH (Supplementary Methods), 0.20 mM cyanocobalamin and 20 mM Ti(III) citrate (or 5 mM DTT) to a final volume of 20 µl. The reaction was placed in tightly closed 2-ml gas chromatography vials within 20-ml adaptors and incubated for 2 h at 37 °C. Gas from the bottle headspace was analysed in a Clarus 690 system (gas chromatography–flame ionization detector/thermal conductivity detector; PerkinElmer) with a custom-made column circuit (ARNL6743) operated with the TotalChrom v.6.3.4 software (PerkinElmer). The headspace samples were injected using a TurboMatrix 110 autosampler (PerkinElmer) and heated at 45 °C for 15 min before injection. The samples were further separated on a HayeSep column (7′ HayeSep N 1/8″ Sf; PerkinElmer), followed by a molecular sieve (9′ Molecular Sieve 13 × 1/8″ Sf; PerkinElmer) kept at 60 °C. Subsequently, CH$_4$ was detected using both flame ionization detector (250 °C) and thermal conductivity detector (200 °C). For comparison, activity assays lacking the pre-incubation step were assayed manually by injecting 100 µl from the headspace of 2-ml gas chromatography vials in an Agilent 6890 gas chromatography–flame ionization detector system connected to OpenLab CDS software (Agilent Technologies). The column used was an HP-Al$_2$O$_3$/KCl 19091P-K15 (50 m × 0.32 mm; 8 µm; 7′; Agilent Technologies), and the run was kept at 40 °C, 25 psi and 3.8 ml min$^{-1}$ helium. The FID was set at 250 °C under 35 ml min$^{-1}$ hydrogen, 350 ml min$^{-1}$ air and 29 ml min$^{-1}$ helium. In all cases, CH$_4$ quantification was based on linear calibration curves from the samples with known amounts of gas. At least three activity assay replicates were carried out from three independent protein preparations.

### Cryo-EM and grid preparation

For cryo-EM data acquisition, QUANTIFOIL R 1.2/1.3 grids on copper 300 mesh (Quantifoil Micro Tools) were glow discharged for 25 s with a current of 15 mA in a PELCO easiGlow device (Ted Pella), before pipetting 4 µl of 'as isolated' protein (1.2 mg ml$^{-1}$) and automatically plunge grids into liquid ethane using a Vitrobot Mark IV (Thermo Fisher Scientific) with blotting force 7 for 7 s at 4 °C and 100% humidity. To determine the structure of the MCR activation complex after the exogenous addition of ATP (5 mM), purified protein (1.2 mg ml$^{-1}$) was mixed with the CH$_4$ reaction components, except Ti(III) citrate. A final volume of 20 µl was incubated for 30 min at room temperature before plunging 4 µl onto the cryo-EM grids under the conditions described above. Vitrification and handling of the samples were always performed under strict anaerobic (redox-controlled) conditions in a vinyl anaerobic chamber (Coy Laboratory Products) filled with 95% N$_2$ and 5% H$_2$.

### Data collection, processing and model building

Cryo-EM data for both 'as isolated' and ATP-incubated samples were acquired using Smart EPU Software on a TFS Krios G4 Cryo-TEM operating at an accelerating voltage of 300 keV and equipped with a Falcon 4i Direct Electron Detector (Thermo Fisher Scientific) and Selectrics Energy Filter set at a 10 eV slit width. For the 'as isolated' sample (Extended Data Fig. 3), data were acquired at a nominal magnification of ×165,000 corresponding to a calibrated pixel size of 0.732 Å. Images were acquired with an exposure dose of 60 e$^-$ Å$^{-2}$ in counting mode and exported as an electron-event representation (EER) file format.

A non-tilted and 20° pre-tilted dataset containing 9,767 and 7,781 micrographs were acquired, respectively. The entire process was carried out in CryoSPARC[63]. The EER files were fractionated into 60 frames, motion corrected using patch motion correction[64], followed by contrast transfer function (CTF) estimation. Particles were initially picked manually and then subsequently used for training a Topaz model[65] on a small subset of the data containing 1,000 micrographs. For the non-tilted dataset, roughly 1.02 million particles could be extracted and used to generate three ab initio classes. Two ab initio major classes containing 406,130 particles (with the A2 component) and 216,922 particles (without the A2 component) were refined to 3.2 and 3.3 Å, respectively, but showed elongation artefacts because of the preferred orientation of the particles. To overcome the orientation bias detected in the non-tilted dataset, we extracted 534,235 particles from the 20° pre-tilted dataset and 3D-classified them using previously generated ab initio models. Particles from the best classes were combined with particles from the non-tilted dataset (both with and without the A2 component) and used in heterogeneous refinement to generate five more classes. For the maps containing the A2 subunit, two classes with 220,928 and 263,088 particles, respectively, were re-extracted with a box size of 500 pixels. The combined particles were subjected to non-uniform refinement, followed by a reference-based motion correction and another non-uniform refinement, generating a map with a resolution of 2.5 Å (Fourier shell correlation cut-off = 0.143). Similarly, a class without an A2 subunit, consisting of 349,437 particles, could be resolved to a resolution of 2.7 Å. Furthermore, we took advantage of masked local refinement to refine the area containing the three FeS clusters (Extended Data Fig. 5c). The mask was placed on the density map without an A2 subunit, and local refinement yielded a map at a resolution of 2.8 Å. Additionally, for the peripheral subunits, which are dynamic and highly flexible, we used 3D masked classification without alignment to generate the best aligning particles, which were then used for masked local refinement. For the Mmp3 subunit, a mask was placed on the map without A2 and was 3D classified into six classes. The top 4 classes were subjected to local refinement using the same mask, which gave a reconstructed map with a resolution of 3.34 Å. The same steps were repeated to refine the A2 component in the map with the A2 subunit. In total, seven classes were 3D classified, and then the best six classes were locally refined to obtain a map in which the A2 component was refined to a nominal resolution of 3.17 Å.

To resolve the high-resolution cryo-EM structure of the MCR complex incubated with ATP (Extended Data Fig. 4), data were acquired at a magnification of ×215,000, which corresponds to a physical pixel size of 0.573 Å. To counter the preferred orientation already observed in the 'as isolated' dataset, we decided to acquire the data at a pre-tilt of 25°. A total of 30,000 images were acquired with an exposure of $60 \, e^- \, Å^{-2}$ in counting mode and exported as an EER file format. All EER frames were imported to CryoSPARC and fractionated into 60 frames, followed by motion correction and CTF estimation. Micrographs with a CTF below 3.5 Å were discarded, and 28,171 micrographs were used for the final processing. An initial set of particles were manually selected from a subset of 100 micrographs, which were then used as input to train a Topaz model. This was followed by iterative rounds of Topaz training to generate a model that was used to select particles from all micrographs. A total of 0.95 million particles were picked, extracted with a box size of 320 pixels and subjected to generate three ab initio models. Of the three ab initio classes, two good classes containing 279,356 and 299,435 particles that showed the MCR complex bound to the A2 component were combined and subsequently subjected to two rounds of heterogeneous refinement. The remaining 171,180 particles were used for 3D classification without alignment, which gave a clean class of MCR complex with the A2 component (118,247 particles). These particles were re-extracted using a bigger box size of 600 pixels and subjected to non-uniform refinement to yield a map with a global resolution of 2.4 Å. To further enhance the resolution of the reconstructed map, the particles were polished using reference-based motion correction and refined again using non-uniform refinement to generate a map with a resolution of 2.1 Å. The local resolution of the map increased to 1.8 Å, which allowed us to accurately model the core of the complex containing the $F_{430}$ and FeS clusters. Furthermore, to investigate the occupancy of the A2 component in the MCR complex, we performed masked 3D classification by placing a soft mask on A2 and used the class containing 171,180 particles. For this job, seven classes were generated, and all except one class contained the A2 component bound to the MCR complex. The class without the A2 component contained 21,000 particles (8%) and was refined up to a resolution of 2.7 Å. The map of the MCR complex without the A2 subunit was identical to the one previously obtained from the 'as isolated' dataset. To refine the A2 component from the other six classes, we performed a masked local refinement to yield a map with a resolution of 2.45 Å. To further improve the quality of McrC, Mmp7 and Mmp3, we placed a soft mask on these regions using the map that was refined at a resolution of 2.1 Å and performed local refinement. The map could be refined with local resolutions as high as 1.8 Å.

We performed all initial modelling of the MCR complex using ModelAngelo (v.1.0)[66] by taking the protein sequences of *M. maripaludis* JJ (DSM 2067; locus tag MMJJ). Moreover, AlphaFold2 (ref. 67) individual homology models were used to enrich ModelAngelo's output and further fitted into the electron density using UCSF Chimera (v.1.17.3). Coot (v.0.9.8.92) was used to manually rebuild the model followed by iterative real-space refinements in PHENIX (v.1.21-5207). All ligands were imported into Coot and manually fitted to the maps before refinement. All figures were prepared using UCSF ChimeraX (v.1.6.1).

## Molecular size determination and UV–Vis

We performed SEC by injecting 100 µl of purified protein into a Superdex 200 Increase 10/300 GL column (Cytiva), previously equilibrated with buffer A and attached to an UltiMate 3000 HPLC system (Thermo Fisher Scientific) with a diode array detector from which the UV–Vis spectrum was also registered. SEC runs were performed under strict anaerobic conditions inside of a vinyl anaerobic chamber with 95% $N_2$ and 5% $H_2$ (Coy Laboratory Products). Chromeleon v.7.2.10 (Thermo Fisher Scientific) was used to register the UV–Vis spectrum from selected peaks in the sample running at 0.4 ml min$^{-1}$, facilitating further data acquisition. A mixture of proteins from the Gel Filtration Calibration Kit (Cytiva) was used as the standard. Alternatively, a OneMP mass photometer connected to AcquireMP v.2.3 (Refeyn) was used to analyse the molecular weight of the protein complex. Videos were recorded at 1 kHz, with exposure times varying between 0.6 and 0.9 ms, adjusted to maximize camera counts while avoiding saturation. Silicon gaskets were fixed to clean glass slides, and the instrument was calibrated using the NativeMark Protein Standard (Thermo Fisher Scientific) immediately before measurements. To blank and find the focus, 18 µl of buffer A with 20 mM Ti(III) citrate and 5 mM nucleotides (ATP or AMP-PNP) was pipetted into a well, and the focal position was locked using the autofocus function of the instrument. The measurement started after adding 2 µl of 0.3 µM protein onto the gasket well, resulting in a ten-fold dilution of the sample before measurement (two repetitions). Data analysis was conducted using DiscoverMP v.2.3 (Refeyn).

## Mass spectrometry

For protein identification, 15% sodium dodecyl sulfate–polyacrylamide gel electrophoresis bands were cut out and intensively washed twice with a buffer solution containing 60 mM $(NH_4)_2CO_3$, 50% acetonitrile and 30 mM thioglycolic acid (pH 8.2) before adding 100% acetonitrile and drying the samples. The samples were incubated overnight at 30 °C in 5 µg ml$^{-1}$ mass spectrometry-approved trypsin (SERVA), 10% acetonitrile and 50 mM $NH_4HCO_3$. To analyse the post-translational modifications in McrA, 9 µg of protein was diluted in 50 mM $NH_4HCO_3$ with 0.5% sodium lauryl sulfate and then subjected to an SP3 protein

purification protocol following the manufacturer's instructions (GE HealthCare). After washing, the beads were reconstituted in 100 µl of the same digestion buffer, incubated for 6 h at 30 °C and magnetically removed after two extraction steps with 0.1% trifluoroacetic acid (TFA). In all cases, the resulting peptide solutions were purified in a CHROMABOND C18 WP spin column (20 mg; Macherey-Nagel) and eluted with 400 µl of a mixture composed of 50% acetonitrile and 0.1% TFA before finally redissolving in 100 µl of 0.1% TFA. Peptides were then analysed using liquid chromatography–mass spectrometry in an Orbitrap Exploris 480 instrument connected to an UltiMate 3000 RSLCnano system with a nanospray ion source (Thermo Fisher Scientific). We performed peptide separation on a reverse-phased high-performance liquid chromatography column (75 µm × 42 cm) packed with C18 resin (2.4 µm; Dr. Maisch) running in a 45-min gradient (0.15% formic acid to 0.15% formic acid + 50% acetonitrile). The mass spectrometry data were analysed using Byonic (Protein Metrics by Dotmatics) and interpreted using Proteome Discoverer 1.4 (Thermo Fisher Scientific) on the basis of the protein sequence information of *M. maripaludis* JJ (DSM 2067).

For determination of metal ions, inductively coupled plasma–triple quadrupole–mass spectrometry (ICP–QQQ–MS) was carried out. In brief, purified and desalted protein samples were subjected to acid digestion by incubating them in 11% (v/v) $HNO_3$ (Suprapur grade; Merck Millipore) for 3 h at 80 °C. Subsequently, the samples were diluted with ultra-pure water to obtain a final $HNO_3$ concentration of 2% (v/v). Calibration standards ranging from 0.005 to 500 µg l$^{-1}$ were prepared by serially diluting the ICP multi element standard solution XVI (Merck Millipore) in 2% (v/v) $HNO_3$. A rhodium internal standard solution was added to all the samples, resulting in a final concentration of 1 µg l$^{-1}$. We analysed the samples using a high-resolution ICP–QQQ–MS system Agilent 8800 (Agilent Technologies) in direct infusion mode with an integrated autosampler. The injection system consisted of a Peltier-cooled (2 °C) Scott-type spray chamber with a perfluoroalkoxyalkane nebulizer, operating at a speed of 0.3 rps for 45 s, using an internal tube diameter of 1.02 mm. Different metals were simultaneously quantified from the Merck XVI standard solution. To mitigate polyatomic interferences, we used an Octopole Reaction System (ORS3) with a collision/reaction cell. Helium and hydrogen were introduced into the collision/reaction cell as collision and reaction gases, respectively, at flow rates of 2.5 and 0.5 ml min$^{-1}$, respectively. The carrier gas (argon) maintained a flow rate of 2.7 ml min$^{-1}$. For each metal, both the first (Q1) and second (Q2) quadrupoles were set to the same $m/z$ value with an integration time of 1 s in an autodetector mode. All measurements were normalized using the internal standard. Furthermore, all other parameters were optimized using the autotune function in the MassHunter 4.2 operation software (Agilent Technologies).

## Electron paramagnetic resonance

The protein samples were analysed at a concentration of 0.01–0.03 mM (approximately 90 µl). Incubation with 0.6 mM dithionite or 0.1 mM thionine was carried out in an anaerobic box (100% $N_2$). The experimental data were recorded on a Bruker EMXplus X-Band spectrometer with an ER355 4122 super-high Q resonator installed. An Oxford ITC4 temperature controller was used to control the temperature of an ESR900 helium flow cryostat (Oxford Instruments). Because of the very low concentration of the samples, a baseline correction was thoroughly performed. Therefore, a reference spectrum obtained from buffer A recorded with identical experimental parameters was subtracted from the spectrum of the sample. Additionally, a spline function was used to correct broad baseline drifts. The experimental parameters used were a microwave frequency of 9.3 GHz, a modulation amplitude of 10 G and a modulation frequency of 100 kHz.

## Phylogenetic analysis

Sequences for McrC, Mmp7, CfbD and the Nif reductases family were collected from the National Center for Biotechnology Information

BLASTP and aligned using Multiple Alignment using Fast Fourier Transform (MAFFT)[68]. For the Nif reductases tree, we searched for homologues of the molybdenum nitrogenase NifK/D and their maturases NifE/N, group IV nitrogenases (including MarK/D), group VI nitrogenases and maturases of the vanadium nitrogenase (VnfE/N). As our goal was to resolve the relationships among the early branching, plausibly L-cluster-using members of this family, we excluded protochlorophyllide reductases and related enzymes (which reduce tetrapyrrole compounds and do not contain L-clusters[69]). We did not consider iron or vanadium nitrogenase catalytic subunits (AnfK/D and VnfK/D, respectively) because it has been established that they descend from molybdenum nitrogenases[60,70]. Thus, we obtained a more manageable dataset for multiple sequence alignment. For NifB, we collected sequences using BLASTP from the National Center for Biotechnology Information and supplemented them with sequences found by blasting versus the Genome Taxonomy Database, specifically within the archaeal taxa. We decided to include the group VI nitrogenases, although there is debate over whether they contain an active site cluster. This makes our inference conservative with respect to the earliest possible utilization of L-clusters in this family, making it possible for group VI nitrogenases to secondarily lose such a cofactor. We decided to infer the Nif reductases tree separately from a CfbD tree, although they are part of the same family, and the root of the combined tree probably lies within CfbD (see section 'MCR activation and nitrogenase evolution'). Given that CfbD (a homodimer) is structurally different from all other family members (heterotetramers), it would have been difficult to align it with other members of the family. The combined alignment of CfbD, Nif and Nif-like reductases was trimmed using trimAl[71] with the 'strict' option, which resulted in only 143 positions and was therefore too short to consider deep relationships with high confidence. Even if this highly trimmed alignment was used, the branch separating CfbD from all heterotetrameric members of this family would be considerably long (more than one), making its placement highly unreliable within CfbD and Nif/Nif-like paralogues. For this reason, we inferred the CfbD phylogeny separately and rooted it between TACK and Euryarchaeota, according to the archaeal species phylogeny[42–44]. The Nif and Nif-like reductase tree was then rooted such that the deepest branch separates all D or D-like paralogues from all K or K-like paralogues. This minimizes the number of gene and operon duplications as well as gene losses relative to roots within both the K and D clades. Lineage-specific insertions were trimmed out of McrC, Mmp7 and CfbD manually because their alignments contained only a few gaps. Because of the different lengths of the Nif sequences, this tree was more challenging to trim. Therefore, we again used trimAl with the 'strict' option for the first trim, before manually trimming the termini of the obtained alignment. Gene trees were inferred using IQ-TREE[72]. The best-fit models were inferred using the in-built ModelFinder[73] routine (-m MFP) of IQ-TREE, using Bayesian information criterion (BIC) as the model choice criterion for McrC, Mmp7, CfbD and the Nif tree; for NifB, Akaike information criterion (AIC) was chosen instead. Statistical support was assessed using ultrafast bootstrap[74] (1,000 for McrC, Mmp7, CfbD and Nif reductases; 10,000 for NifB).

## Reporting summary

Further information on research design is available in the Nature Portfolio Reporting Summary linked to this article.

## Data availability

The cryo-EM maps reported in this article are available in the Electron Microscopy Data Bank with the accession codes EMD-51767 (MCR activation complex + A2 after incubation with ATP), EMD-19787 (MCR activation complex as isolated + A2) and EMD-19788 (MCR activation complex as isolated without A2). The local refined mask maps for A2, FeS containing subunits and Mmp3 were submitted along with EMD-51767

and EMD-19787. The atomic models are deposited in the Protein Data Bank under the codes 9H1L (MCR activation complex + A2 after incubation with ATP), 8S7V (MCR activation complex as isolated + A2) and 8S7X (MCR activation complex as isolated without A2). Structural data used for comparison are available in the Protein Data Bank under the codes 3M32 (MCR from *M. marburgensis* bound to CoMS–SCoB), 1HBM (MCR from *Methanothermobacter thermautotrophicus* bound to CoMS–SCoB), 8CRS (cryo-EM structure of Mo-nitrogenase from *Azotobacter vinelandii*), 8DPN (cryo-EM structure of Mo-nitrogenase from *A. vinelandii* DJ), 6FEA (V-nitrogenase from *A. vinelandii*), 8OIE (cryo-EM structure of Fe-nitrogenase from *Rhodobacter capsulatus*), 8BOQ (Fe-nitrogenase from *A. vinelandii*), 3U7Q (Mo-nitrogenase from *A. vinelandii* at atomic resolution), 3PDI (precursor-bound NifEN from *A. vinelandii* DJ) and 7BI7 (P-cluster-bound NifB from *M. thermautotrophicus*). Sequences for comparison with the A2 component are available in UniProt with accession numbers A0A8T3W5C7, D9PXN7, D7DTW8, Q7LYR5, Q58639, Q8TIZ1, A0A0E3R7T8, A0A0E3N9D1, F8ANN2, A6USB5, A0A8J8F9M9 and A0A8S9VY72. Original sequence alignments and phylogenetic trees have been deposited into Edmond, the Open Research Data Repository of the Max Planck Society, for public access and available at https://doi.org/10.17617/3.UZSFCW. All data needed to evaluate the conclusions in the paper are present in the paper and/or the Supplementary Information. Source data are provided with this paper.

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

**Acknowledgements** We acknowledge the contributions of the Cryo-EM Facility and the Protein Biochemistry and Spectroscopy Facility of the Philipps-University Marburg. We thank N. Oehlmann, J. Goenrich, E. Zimmer and J. Bao for their technical assistance, and C. Gabel for critical reading of the paper. We are grateful to R. Thauer for constructive discussions and insights. D.D. acknowledges the Department of Environmental Analytical Chemistry at UFZ-Leipzig for enabling metal quantification through ICP–QQQ–MS. J.M.S. acknowledges the Deutsche Forschungsgemeinschaft for an Emmy Noether grant (SCHU 3364/1-1), RTG 2937 and the European Union's Horizon 2020 research and innovation programme (Two-CO2-One; grant agreement no. 101075992). The views and opinions expressed are those of the author(s) only and do not necessarily reflect those of the European Union or the European Research Council. Neither the European Union nor the granting authority can be held responsible for them. F.R.-A. acknowledges the funding from the International Max Planck Research School Principles of Microbial Life. S.S. thanks the Novo Nordisk Foundation (grant no. NNF19OC0054329). C.L. was supported by the Deutsche Forschungsgemeinschaft (German Research Foundation) through the cluster of excellence 'UniSysCat' under Excellence Strategy (EXC2008-390540038).

**Author contributions** J.M.S. conceptualized the project. J.M.S., S.S., S.T.S. and F.R.-A. coordinated the experiments. F.R.-A. and S.P. designed and performed cloning, cultured cells and expressed and purified the proteins. A.K. and S.B. collected the cryo-EM data. F.R.-A. processed the initial data and built refined models. A.K. optimized the data processing, performed local refinements and refined the final models. F.R.-A., C.L. and S.T.S. conducted spectroscopic analyses. F.R.-A., S.K., T.N. and S.S. designed and performed enzymatic assays. S.L., D.V. and G.H. constructed phylogenetic trees. J.K. and D.D. carried out mass spectrometry experiments. F.A. and O.V. chemically synthesized coenzyme B. F.R.-A., S.P., A.K., C.L., S.S., S.T.S. and J.M.S. analysed and interpreted the experimental data. F.R.-A. and J.M.S. wrote the paper, with insights from all other authors. F.R.-A., S.P. and A.K. contributed equally to this work.

**Funding** Open access funding provided by Philipps-Universität Marburg.

**Competing interests** The authors declare no competing interests.

**Additional information**
**Correspondence and requests for materials** should be addressed to Jan Michael Schuller.

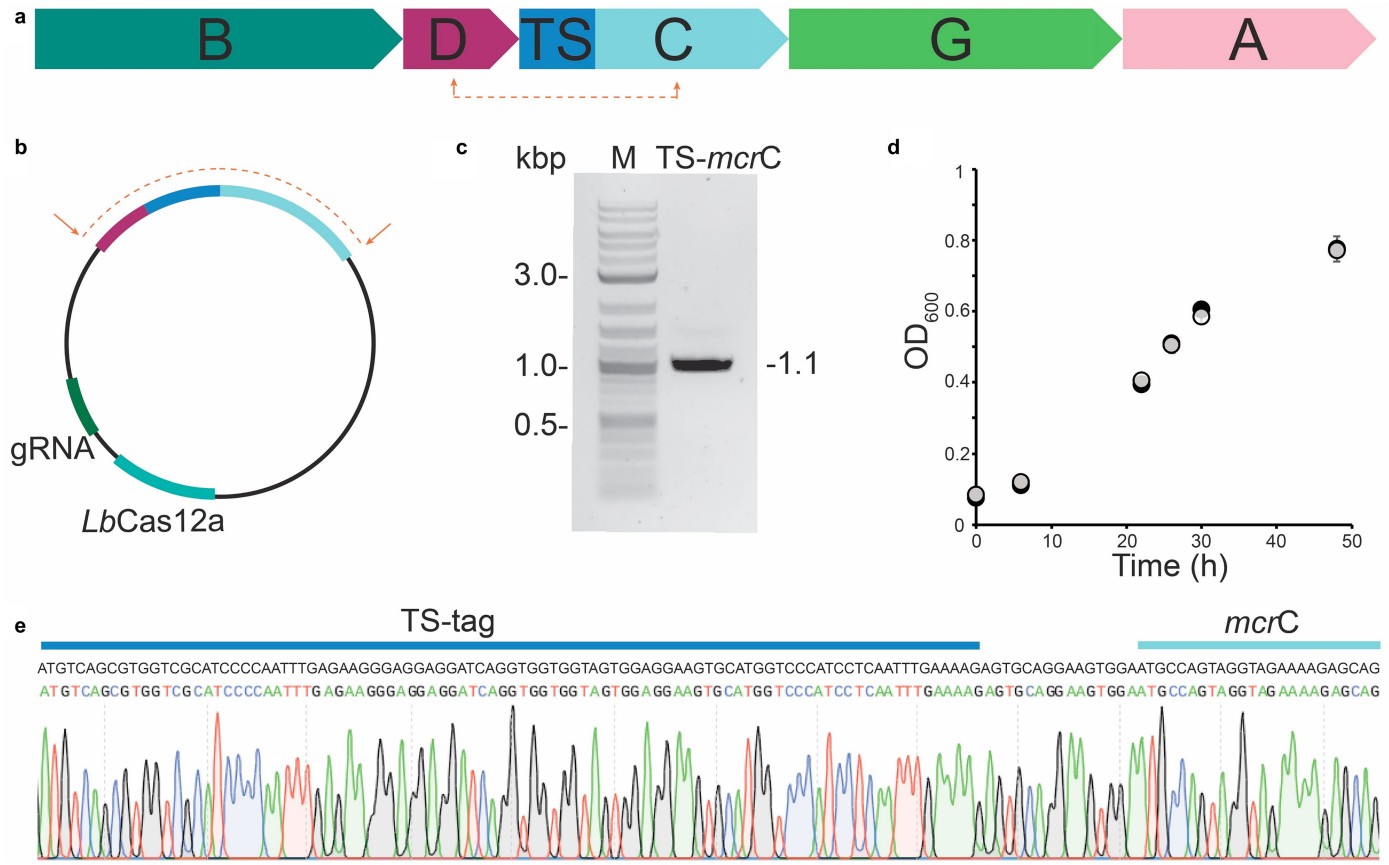

ATGTCAGCGTGGTCGCATCCCCAATTTGAGAAGGGAGGAGGATCAGGTGGTGGTAGTGGAGGAAGTGCATGGTCCCATCCTCAATTTGAAAAGAGTGCAGGAAGTGGAATGCCAGTAGGTAGAAAAGAGCAG

**Extended Data Fig. 1 | Integration of DNA sequence encoding for N-terminal Twin-Strep (TS) tag in the genome of *M. maripaludis*. a**, *mcr* operon in *M. maripaludis*. The N-terminal TS-tag was inserted before *mcrC*. **b**, Plasmid map of pMM002p/TS-*mcrC* (see Supplementary Table 2). The dashed line shows the homology overhangs covering 750 bp of the upstream and downstream sequences from the TS insertion site via the CRISPR/Cas12a system (see Methods). **c**, PCR confirmation of the TS-tag integration before *mcrC* in 1% agarose gel. At least three independent gels showed the same separation pattern (for uncropped gel see Supplementary Fig. 1). **d**, Growth curve of *M. maripaludis* JJ Δ*upt* (black circles) and cells carrying TS-*mcrC* (white circles). Datapoints are the mean ± s.e.m of two independent biological replicates, measured in duplicates. **e**, Sanger sequencing with genome-specific primers (see Supplementary Table 1).

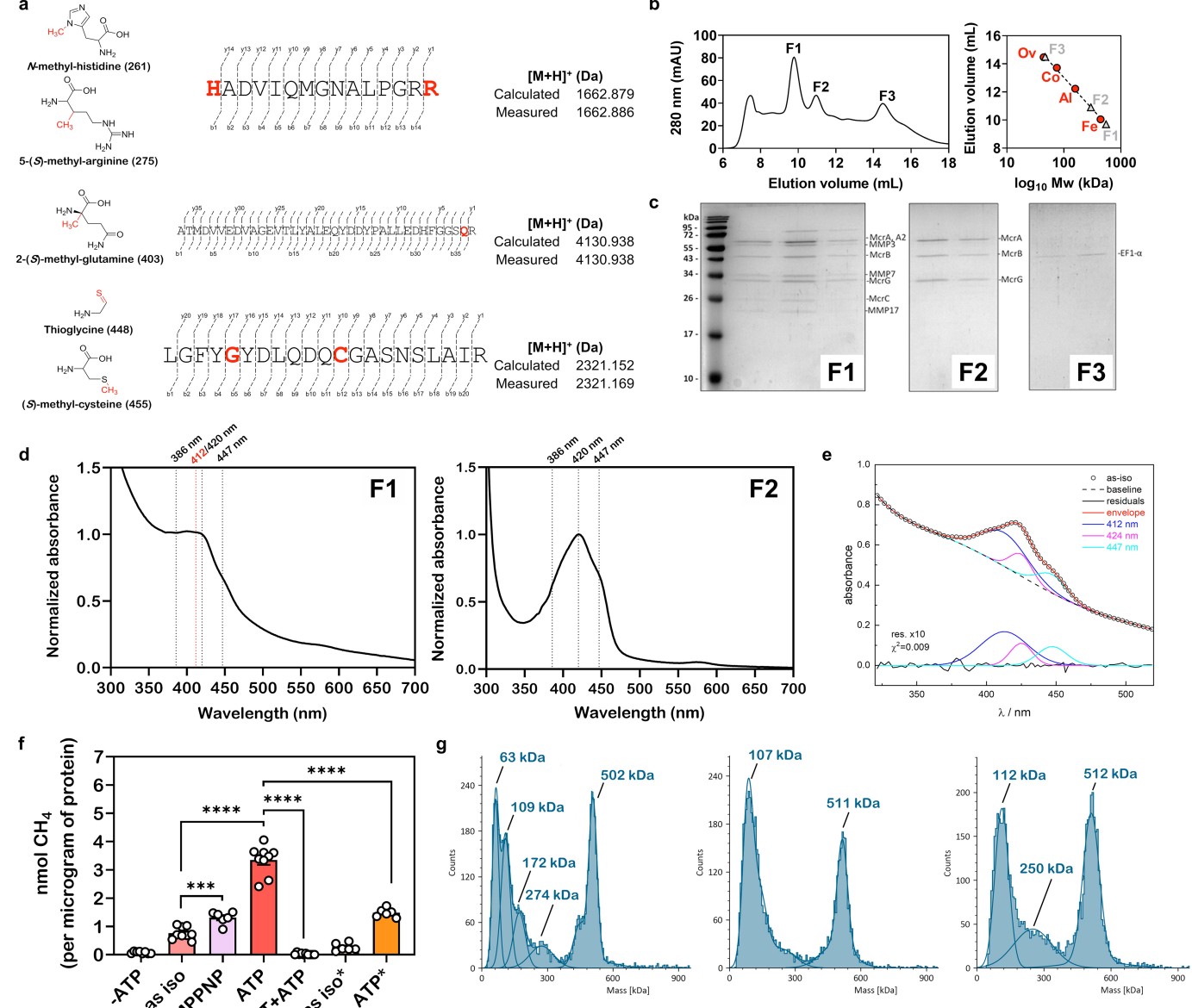

**Extended Data Fig. 2 | Characterization of the MCR activation complex.**
**a**, MS analysis of post-translational modifications in McrA. *Left*: structures of the amino acid modifications (residue number in McrA). *Middle*: peptide fragmentation. *Right*: daughter ion assignments for digested peptides. mHis261 and mArg275 were found in a unique peptide, as well as tGly448 and mCys455. **b**, Fractions obtained by SEC and calibration curve. Standard proteins are shown as red circles [ovalbumin (Ov), 44 kDa; conalbumin (Co), 75 kDa; aldolase (Al), 158 kDa; ferritin (Fe), 440 kDa] and the sample fractions as gray triangles [EF1-α (F3), 47 kDa; MCR (F2), 298 kDa; MCR + activation complex (F1), 551 kDa – out of calibration limit]. **c**, SDS-gel with fractions obtained from **b** (for uncropped gel see Supplementary Fig. 1). At least two gels were tested for confirmation. **d**, UV-Vis absorption spectra of MCR + activation complex (F1) and MCR core (F2); see **b** and **c**. Dashed lines denote wavelengths at 386, 412 (red), 420 and 447 nm. At least three independent biological replicates were carried out, taking one representative to plot. **e**, Electronic spectrum of the 'as isolated' MCR + activation complex (F1). Data fitted with a minimum of three Voigt functions and a single-order polynomial "baseline", $\chi^2 = 0.009$ (residuals are multiplied x10 for visualization). The spectrum agrees with a broad

component at 412 nm (blue, FWHM = 40) and more narrow components at 424 nm (magenta, FWHM = 18) and 447 nm (cyan, FWHM = 24). The envelope (red) accounts for the sum of these three bands and the polynomial baseline. All fits were performed with Global Gauss Fit v7.0 (courtesy of Dr. Petko Chernev, Uppsala University). Tentatively, the highest energy band is assigned to an oxidized iron-sulfur cofactor[75], i.e., similar to $[4Fe\text{-}4S]^{+2}$. The bands at lower energies likely represent the $F_{430}$ cofactor in the Ni(II) state (424 nm and 447 nm, $MCR_{silent}$)[76]. Contributions from Ni(I)-$F_{430}$ (386 nm, $MCR_{red1}$) cannot be excluded but did not increase the fit quality. **f**, Activity assays showing $CH_4$ per µg of total protein including complementary experiments [(AMP-PNP, DTT instead of Ti(III)-citrate, and without pre-activation (*, see Methods)]. All data values are shown as mean ± s.e.m. (error bars) from two (AMP-PNP, not pre-activated) or three (DTT) independent biological replicates, with three technical replicates each. Two-sided unpaired *t*-tests performed to compare the reactions with AMP-PNP vs 'as iso' ($p = 0.004$); ATP vs 'as iso' ($p < 0.0001$); ATP vs DTT ($p < 0.0001$) and ATP vs ATP* ($p < 0.0001$). **g**, Mass photometry histograms of MCR+activation complex 'as isolated' (*left*), and after incubation with 20 mM Ti(III)-citrate plus 5 mM ATP (*middle*) or 5 mM AMP-PNP (*right*).

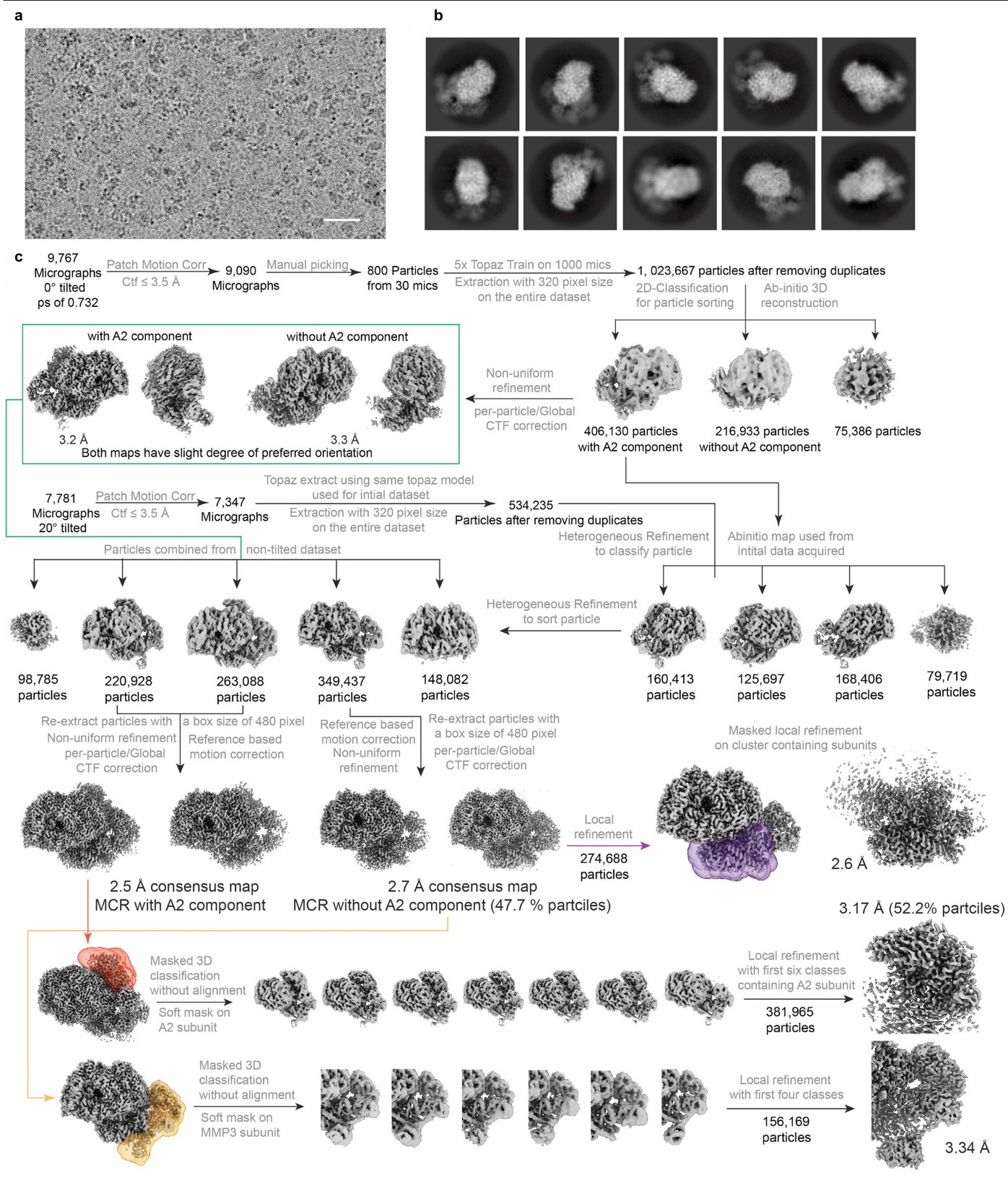

**Extended Data Fig. 3 | CryoEM data processing workflow of MCR + activation complex 'as isolated'. a**, Representative micrograph (total micrographs = 17,548; scale bar = 50 nm) showing the MCR activation complex vitrified under anaerobic conditions (see Methods). **b**, Reference-free 2D classes from combined (non- and 20° tilted datasets) of MCR and its activation complex showcasing different orientations. **c**, Processing tree for determining the structure of the MCR activation complex and its peripheral parts (see Methods).

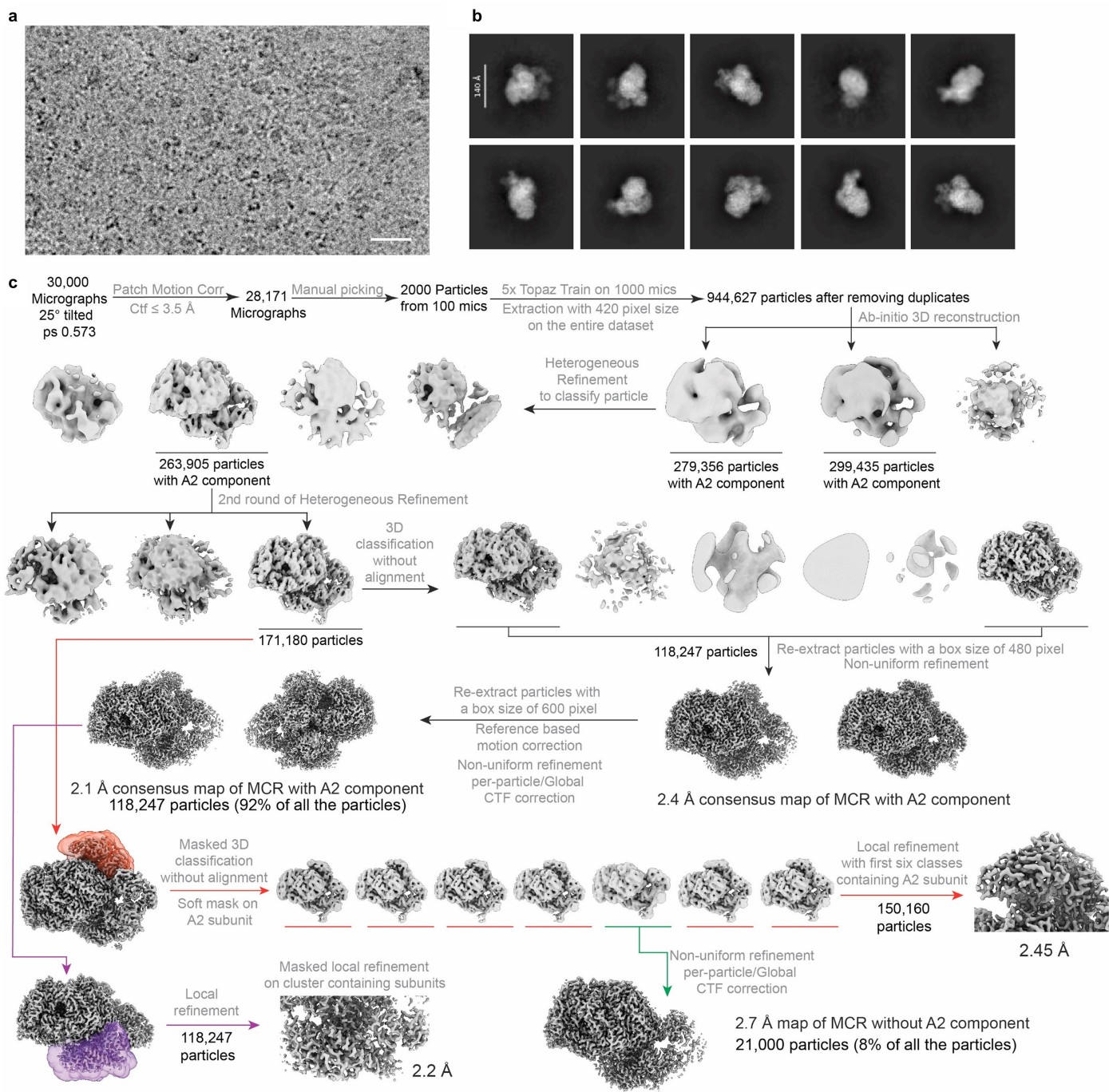

**Extended Data Fig. 4 | CryoEM data processing workflow after incubating sample with ATP. a**, Representative micrograph (total micrographs = 30,000; scale bar = 25 nm) showing the MCR activation complex vitrified under anaerobic conditions in the presence of ATP (see Methods). **b**, Reference-free 2D classes from a 25° tilted dataset of MCR and its activation complex showcasing different orientations. **c**, Processing tree for determining the high-resolution structure of the MCR activation complex and its peripheral parts (see Methods).

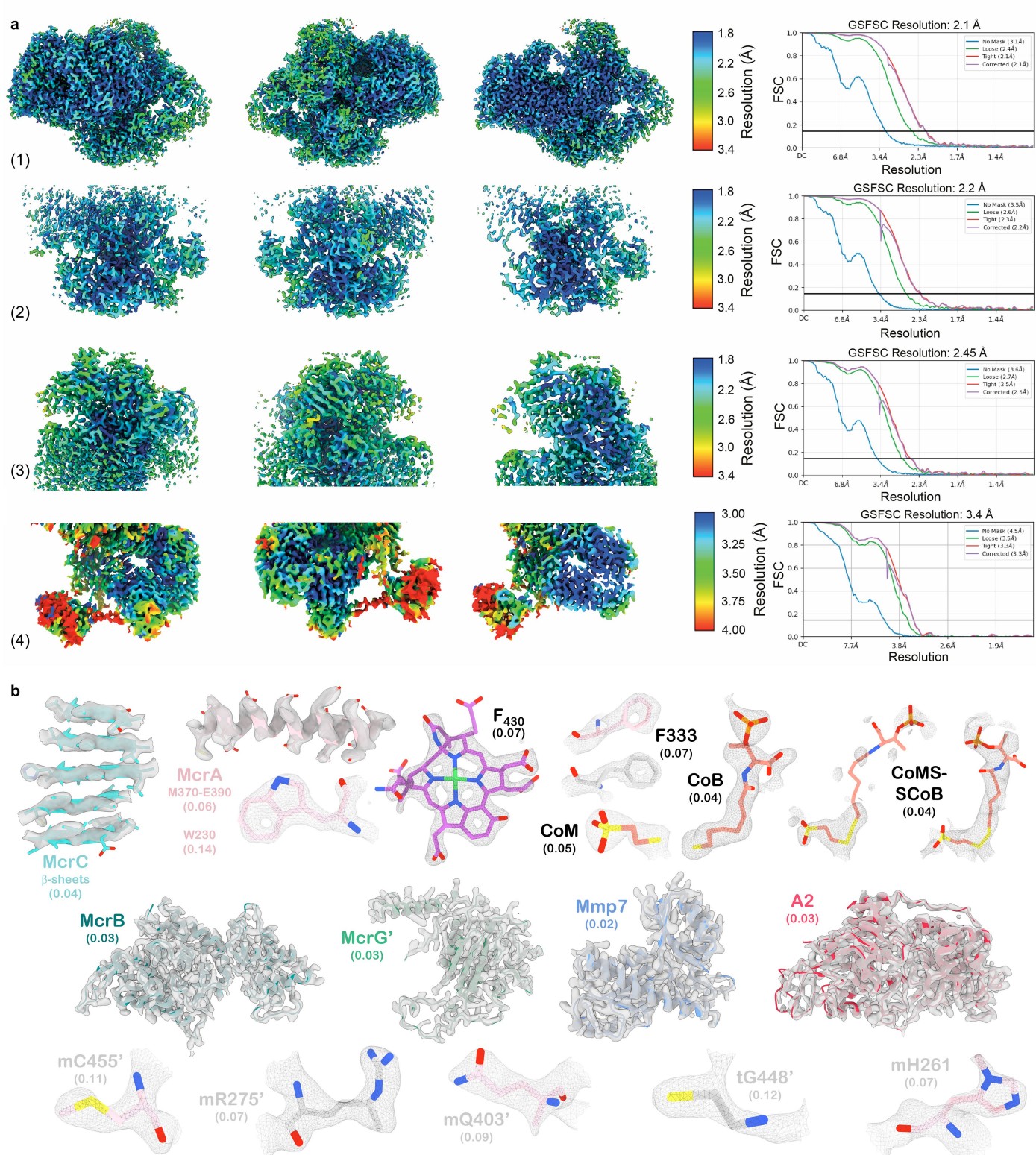

**Extended Data Fig. 5 | CryoEM maps resolution and quality. a**, CryoEM maps colored by their local resolution calculated by GSFSC (plots). Two orthogonal views (*left* and *middle*) and a central cut-open view (*right*) are shown for (1) the complex with the A2 component, as well as the locally refined regions for (2) subunits carrying FeS clusters, (3) the A2 component and (4) Mmp3. (1–3) correspond to the dataset after the incubation with ATP, whereas (4) was obtained from the 'as isolated' sample. Lower resolution in the N-terminal domain of Mmp3 suggests increased motion. **b**, Electron-density maps of representative domains, secondary structure, ligands and post-translational modifications. The contour level is shown between brackets. The density map of the flexible CoMS-SCoB is shown for both the sample incubated with ATP (*left*) and 'as isolated' (*right*).

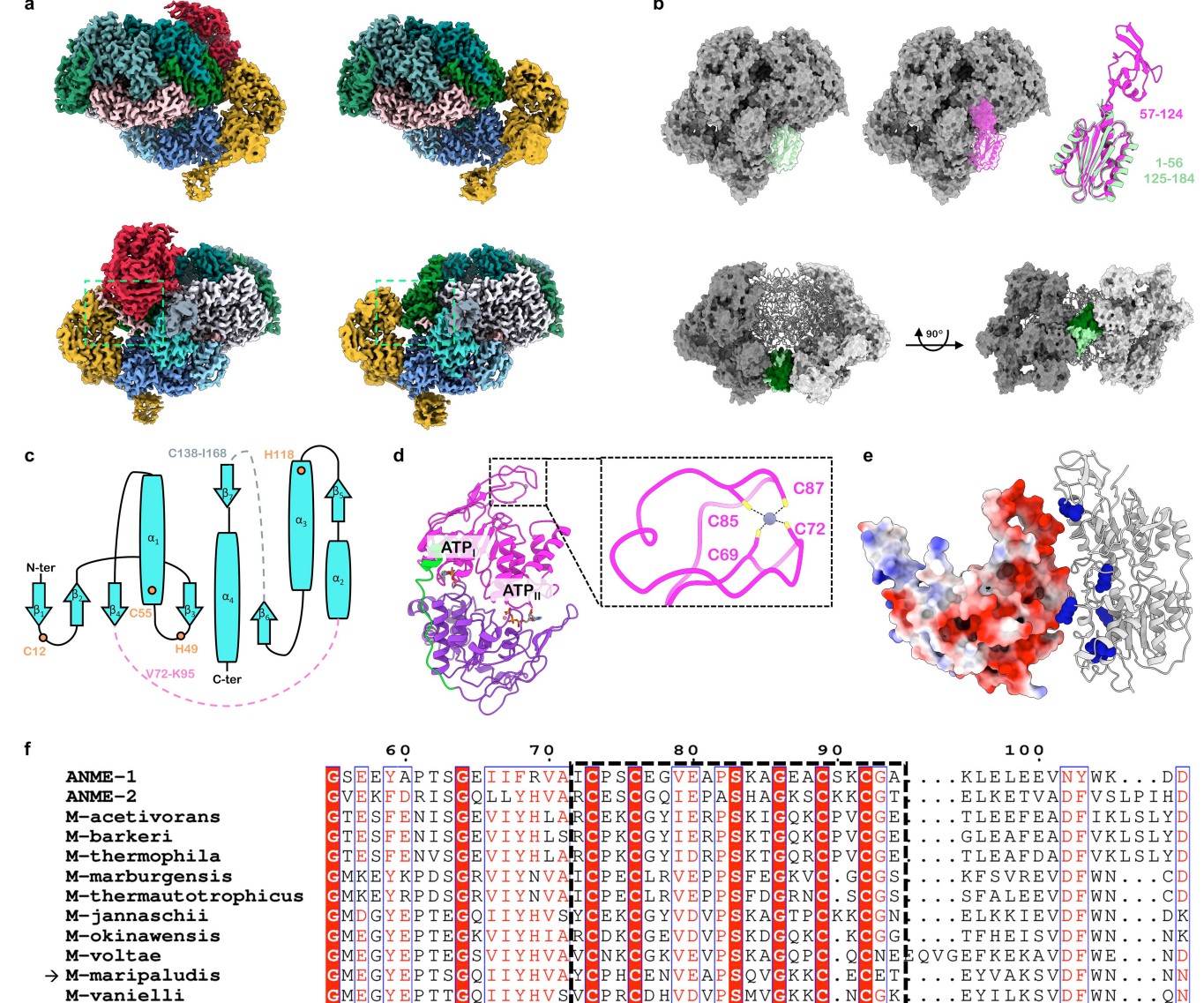

**Extended Data Fig. 6 | Main features of the MCR activation complex components. a**, CryoEM segmented map of MCR's activation complex 'as isolated' with (*left*) and without (*right*) the A2 subunit. The disordered N-terminal region of McrA is enclosed in a square. **b**, *Top:* incomplete Mmp17 domain (green) as obtained from the electron-density maps and AlphaFold2 model of full Mmp17 (magenta, Uniprot A0A2L1C8U1). RMSD = 1.15 Å when aligned. The rest of the model is shown in dark gray surface. *Bottom:* front and bottom views of two overlayed sets of the activation complex (clear and dark gray surfaces) binding upon the MCR core (cartoon). The overlapping Mmp17 subunits (clear and dark green) avoid the simultaneous binding of the activation machinery on both halves of MCR. **c**, Secondary structure topology of the McrC domain. The α-helices and β-sheets are depicted as squares and arrows, respectively. Dashed lines represent loops interacting with McrA (pink) and

DUF2098 (gray). Residues coordinating FeS clusters are shown as orange dots. **d**, Structure of the A2 component, composed by two antiparallel ABC domains (magenta and purple) joined together by a linker (green). Zn is depicted as a gray sphere. The dashed square is a zoom-in of the Zn-binding motif with coordinating cysteines depicted as sticks. **e**, Electrostatic interactions between the A2 component (cartoon) and the proximal McrG (surface). The positively charged residues from A2 are shown as blue spheres, while McrG is colored from the more negative (red) to the more positive (blue) charges. **f**, Multiple-sequence-alignment (MSA) showing conservation of the Zn-binding motif (dashed square) in the A2 component of methanotrophic (ANME-1, ANME-2) and methanogenic species (all others). The arrow indicates the model methanogen utilized in this study (*M. maripaludis*). The MSA was built in MUSCLE and rendered with ESPript 3.0.

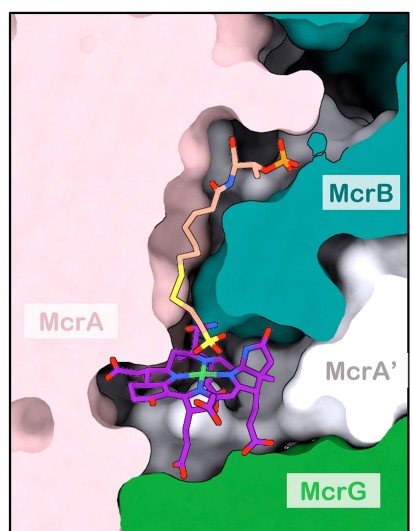

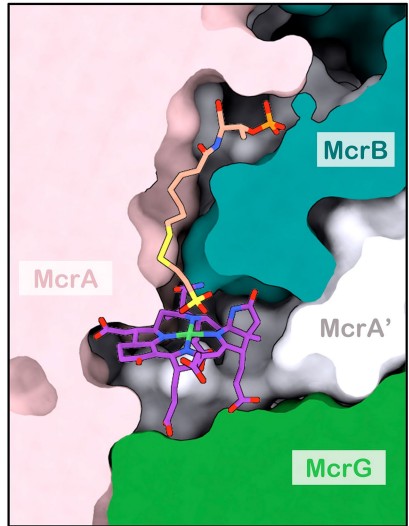

### PDB ID: 3M32

### PDB ID: 1HBM

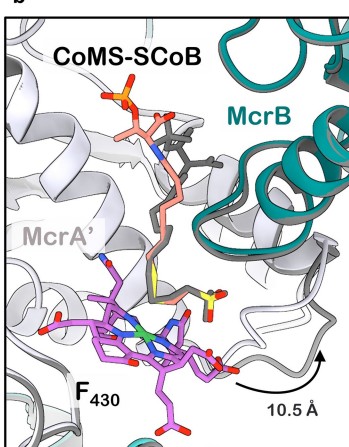

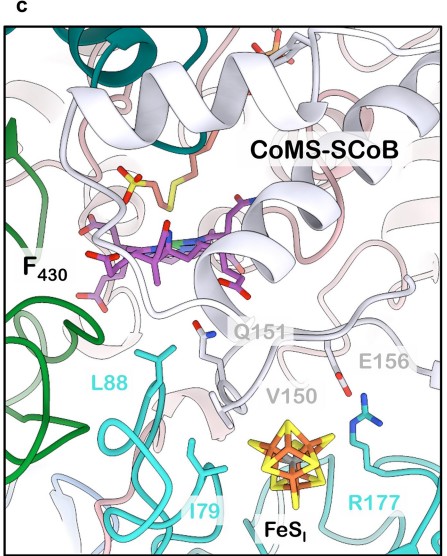

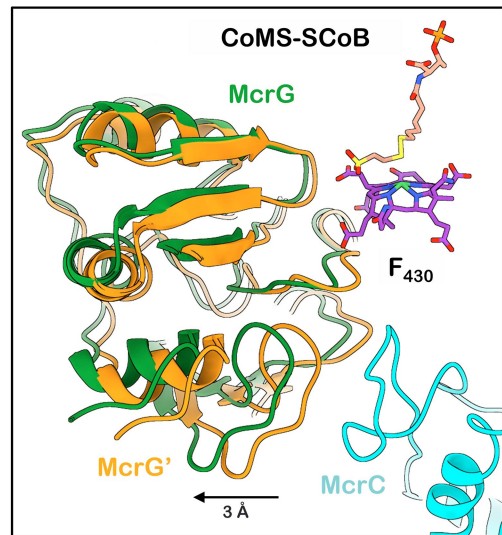

**Extended Data Fig. 7 | Conformational changes in the active site of MCR binding to the activation complex. a**, Close-up of the active sites in X-ray crystallographic structures of MCR showing an alternative product-bound state[3,24]. **b**, Superposition of models of the protein vitrified 'as isolated' (gray) and after incubating with ATP. High flexibility of the CoB moiety in CoMS-SCoB and the K244-E249$^{McrA'}$ loop are shown. **c**, Zoom-in of McrC wedging towards McrA'. Salt bridges between McrA' (E156) and McrC (R177) stabilize the opening. **d**, Superposition of distal (orange) and proximal McrG subunits showing the displacement caused by the interactions with McrC.

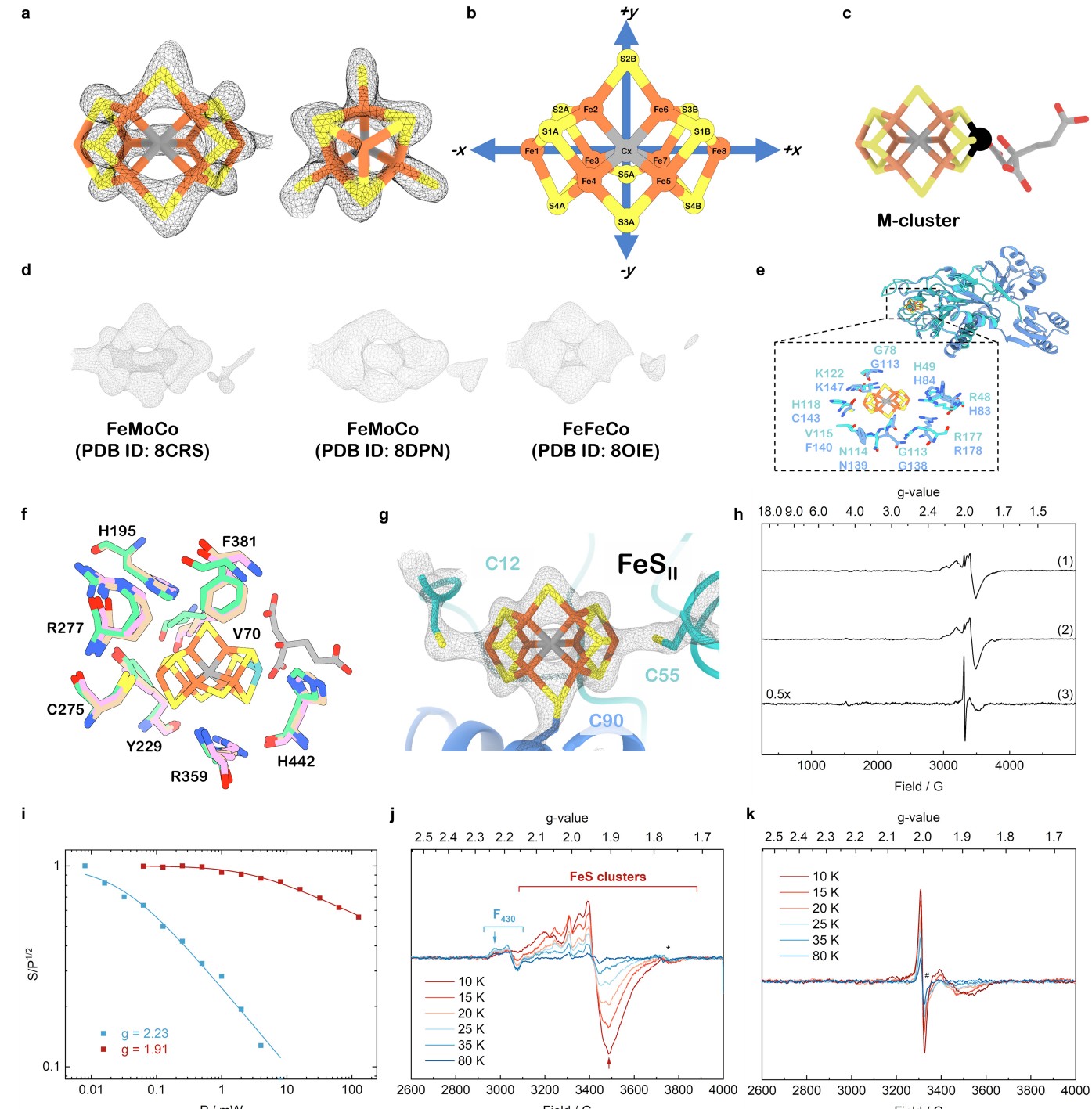

**a**

**b**

**c**

M-cluster

**d**

FeMoCo
(PDB ID: 8CRS)

FeMoCo
(PDB ID: 8DPN)

FeFeCo
(PDB ID: 8OIE)

**e**

**f**

**g**

FeS$_{II}$

**h**

**i**

**j**

**k**

**Extended Data Fig. 8** | See next page for caption.

**Extended Data Fig. 8 | Topological analysis and EPR of the FeS clusters in the activation complex. a**, Different views of the FeS clusters found in the activation complex of MCR. The electron-density clouds were obtained at a contour level of 0.23 σ. **b**, Structure of an L-cluster depicted as a coordinate system. Modified from[35]. **c** and **d**, Structure and electron-densities of the M-cluster, respectively, taken from cryoEM resolved structures of nitrogenases[30–32]. The homocitrate moiety is shown in gray sticks and the black sphere represents molybdenum, vanadium or iron. **e**, Superposition of McrC (cyan) and the McrC-like domain of Mmp7 (blue, RMSD = 2.5 Å). For clarity, only the amino acid residues composing the second coordination sphere of $FeS_I$ (McrC) and $FeS_{III}$ (Mmp7) are shown. **f**, Amino acid residues composing the second coordination sphere around the M-cluster in the molybdenum- (green, 8CRS), vanadium- (pink, 6FEA) and iron-only nitrogenases (tan, 8OIE). The ligand and residue numbering correspond to those in the Mo-nitrogenase from *A. vinelandii*. **g**, Alternative coordination of the $FeS_{II}$ cluster by $C90^{Mmp7}$ possibly bridging between Fe3 and Fe7. The side chain of $Cys90^{Mmp7}$ was reoriented manually and S5A was deleted for visualization. Contour level 0.2 σ. **h**, EPR wide-field scans of the as-isolated (1), sodium dithionite (2) and thionine (3) treated samples recorded at 10 K. **i**, Power-dependent saturation curves of the EPR signals from $F_{430}$ and the FeS clusters recorded at 15 K between 8 μW and 126 mW. The normalized experimental data were fitted using the empirical relation from Portis and Castner[77–79], describing the expected power saturation behavior. S is the signal intensity, $P_{1/2}$ is the power of half-saturation and b corresponds to inhomogeneous (b ≈ 1) or homogenous (1 > b > 3) broadening of the signals. A value for b < 1 is characteristic for dipolar interaction with a paramagnetic site in the vicinity[80]. The paramagnetic nickel species (g = 2.23) was saturated over the entire monitored power range. The power saturation curve from the FeS cluster signal at g = 1.91 clearly shows a decreased saturation, which indicates an enhanced spin relaxation rate, suggesting a dipolar coupling. This is confirmed by the calculated b-value of 0.3. **j** and **k**, Temperature-dependent EPR spectra of the complex incubated with sodium dithionite and thionine, respectively. The weak signal at g = 1.77 (*) in the sample reduced with dithionite (**j**) does not exhibit an unambiguous temperature dependence and could not be assigned to any specific species. The field positions for the power saturation experiments (**i**) are labelled with arrows. The sharp line at g = 2.003 in **k** (#) is presumably related to a minor population of an organic radical formed during thionine treatment.

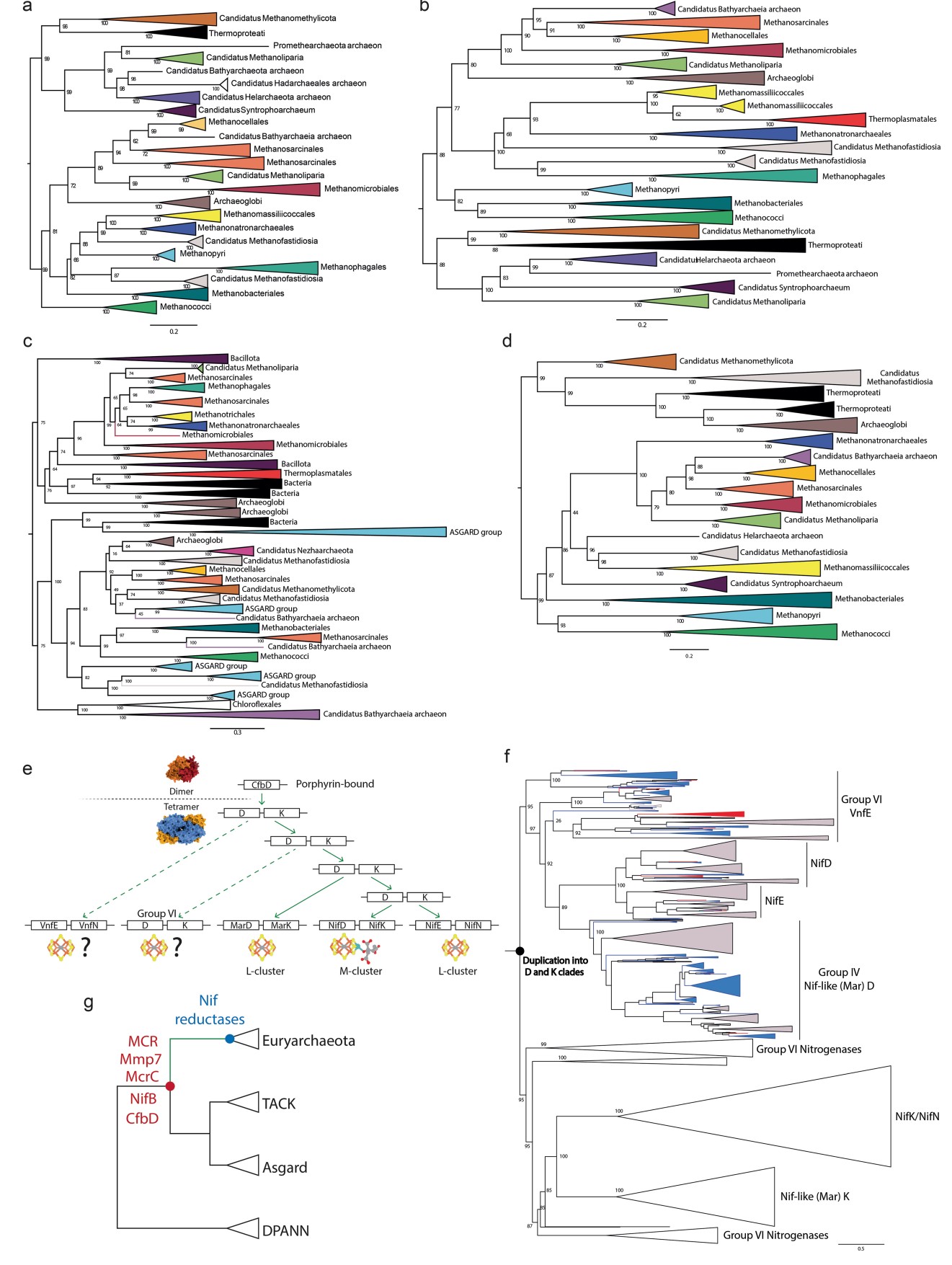

**Extended Data Fig. 9** | See next page for caption.

**Extended Data Fig. 9 | Maximum likelihood phylogenetic trees of McrC (a), Mmp7 (b), NifB (c), CfbD (d) and phylogeny of Nif members (e, f, g).** Trees **a, b** and **d** were rooted between Proteoarchaeota [including TACK archaea (Thermoproteati, Methanomethylicota and Bathyarchaeota)], Asgard archaea (Helarchaeota) and Euryarchaeota. The Proteoarchaeota group contains several Euryarchaeal sequences (Syntropharchaeum, Methanoliparia, Archaeoglobi and Methanofastidiosa), which are horizontal acquisitions also observed in the phylogeny of MCR[81]. In **c**, the phylogeny of NifB was arbitrarily rooted. The root was placed between one group containing most Proteoarchaeota sequences and another containing most Euryarchaeota sequences, implying several horizontal transfer events from Proteoarchaeaota to Euryarchaeota, and from Proteoarchaeota to Bacteria or vice versa. The exact root position is unclear on this tree, but NifB is clearly widely distributed in TACK, Asgard, and Euryarchaeaota. Ultrafast boostrap are indicated at the key nodes. **e**, Gene duplication events from the homodimeric ancestor CfbD. CfbD is the only homodimeric member of the Nif family, likely representing the outgroup to all other heterotetrameric members comprising two different paralogs (usually designated D and K for complexes that function as reductases, or E and N for maturase complexes). The exact order of divergence between group VI and VnfE/N is unresolved (dashed arrows) and whether either group binds L- or M- clusters is not currently known. **f**, Reconstructed phylogenetic tree of Nif proteins. This tree was rooted between the D and K paralogs. The D clade is expanded to show Proteoarchaeota (TACK/Asgard, red), Euryarchaeota (blue) and Bacteria (gray) sequences within each paralog. Proteoarchaeal sequences are very scattered across the tree, implying later horizontal acquisitions of cluster-bearing Nif proteins in this group, rather than an origin in the last common ancestor of Euryarchaeota and Proetoarchaeota. Ultrafast bootstrap support is shown at the key nodes. **g**, Simplified scheme on the origin of L-cluster binding proteins, mapped on a simplified archaeal species phylogeny. MCR and its associated maturation machinery are present in the last common ancestor of TACK, Asgard and Euryarchaeota. L-cluster bearing Nif-reductases only appear in methanogenic members of Euryarchaeota, no earlier than the last common ancestor of this group.

**Extended Data Table 1 | CryoEM data collection, refinement and validation statistics**

| | MCR activation complex + A2 component after incubation with ATP (EMDB-51767) (PDB 9H1L) | MCR activation complex + A2 component (EMDB-19787) (PDB 8S7V) | MCR activation complex without A2 component (EMDB-19788) (PDB 8S7X) |
|---|---|---|---|
| **Data collection and processing** | | | |
| Magnification | 215,000x | 165,000x | 165,000x |
| Voltage (kV) | 300 | 300 | 300 |
| Electron exposure (e–/Å$^2$) | 60 | 60 | 60 |
| Defocus range (μm) | 0.5-2.0 | 0.5-2.0 | 0.5-2.0 |
| Pixel size (Å) | 0.57 | 0.73 | 0.73 |
| Symmetry imposed | C1 | C1 | C1 |
| Initial particle images (no.) | 944,627 | 1,023,667 | 534,235 |
| Final particle images (no.) | 118,247 | 484,016 | 349,437 |
| Map resolution (Å) | 2.17 | 2.56 | 2.78 |
| FSC threshold | 0.143 | 0.143 | 0.143 |
| Map resolution range (Å) | 1.8-3.4 | 2.2-3.2 | 2.2-3.2 |
| | | | |
| **Refinement** | | | |
| Initial model used | *de novo* (ModelAngelo, AlphaFold) | *de novo* (ModelAngelo, AlphaFold) | *de novo* (ModelAngelo, AlphaFold) |
| Model resolution (Å) | 2.17 | 2.56 | 2.78 |
| FSC threshold | 0.5 | 0.5 | 0.5 |
| Model resolution range (Å) | 2.1-2.8 | 2.5-3.1 | 2.7-3.1 |
| Map sharpening *B* factor (Å$^2$) | 45.4 | 73.1 | 100.2 |
| Model composition | | | |
| Non-hydrogen atoms | 32,524 | 32,524 | 28,139 |
| Protein residues | 4,143 | 4,143 | 3,596 |
| Ligands | F43: 2, COM: 1, TP7:1, SHT: 1, S5Q: 3, ATP: 2, MG: 2, ZN: 1 | F43: 2, COM: 1, TP7:1, SHT: 1, S5Q: 3, ATP: 2, MG: 2, ZN: 1 | F43: 2, COM: 1, TP7:1, SHT: 1, S5Q: 3 |
| *B* factors (Å$^2$) | | | |
| Protein | 84.35 | 101.52 | 71.94 |
| Ligand | 71.36 | 112.11 | 78.48 |
| R.m.s. deviations | | | |
| Bond lengths (Å) | 0.003 | 0.003 | 0.003 |
| Bond angles (°) | 0.758 | 0.915 | 0.949 |
| Validation | | | |
| MolProbity score | 2.05 | 2.22 | 2.15 |
| Clashscore | 5.84 | 9.5 | 9.45 |
| Poor rotamers (%) | 3.31 | 3.31 | 2.82 |
| Ramachandran plot | | | |
| Favored (%) | 95.23 | 95.44 | 95.62 |
| Allowed (%) | 4.58 | 4.43 | 4.35 |
| Disallowed (%) | 0.20 | 0.12 | 0.03 |

Processing and refinement data are shown for the different functional states of the MCR activation complex.

# Reporting Summary

## Statistics

For all statistical analyses, confirm that the following items are present in the figure legend, table legend, main text, or Methods section.

| n/a | Confirmed | |
|---|---|---|
| ☐ | ☒ | The exact sample size (*n*) for each experimental group/condition, given as a discrete number and unit of measurement |
| ☐ | ☒ | A statement on whether measurements were taken from distinct samples or whether the same sample was measured repeatedly |
| ☐ | ☒ | The statistical test(s) used AND whether they are one- or two-sided<br>*Only common tests should be described solely by name; describe more complex techniques in the Methods section.* |
| ☒ | ☐ | A description of all covariates tested |
| ☒ | ☐ | A description of any assumptions or corrections, such as tests of normality and adjustment for multiple comparisons |
| ☐ | ☒ | A full description of the statistical parameters including central tendency (e.g. means) or other basic estimates (e.g. regression coefficient) AND variation (e.g. standard deviation) or associated estimates of uncertainty (e.g. confidence intervals) |
| ☐ | ☒ | For null hypothesis testing, the test statistic (e.g. *F*, *t*, *r*) with confidence intervals, effect sizes, degrees of freedom and *P* value noted<br>*Give P values as exact values whenever suitable.* |
| ☒ | ☐ | For Bayesian analysis, information on the choice of priors and Markov chain Monte Carlo settings |
| ☒ | ☐ | For hierarchical and complex designs, identification of the appropriate level for tests and full reporting of outcomes |
| ☒ | ☐ | Estimates of effect sizes (e.g. Cohen's *d*, Pearson's *r*), indicating how they were calculated |

*Our web collection on statistics for biologists contains articles on many of the points above.*

## Software and code

Policy information about availability of computer code

| | |
|---|---|
| Data collection | CryoEM micrographs: Thermo Scientific EPU 3 software; Size exclusion chromatography data: Chromeleon v7.2.1; Mass photometry data: AcquireMP v2.3; Gas chromatography data: OpenLab CDS and TotalChrom v.6.3.4; Mass spectrometry data: Byonic (MS) and MassHunter 4.2 (ICP-MS). |
| Data analysis | CryoEM data: cryoSPARC v4; Protein structure and imaging: UCSF ChimeraX 1.6.1 and UCSF Chimera 1.17.3; Model building and refinement: Coot, PHENIX and eLBOW packages and tools; Prediction of structural models: AlphaFold 2and ModelAngelo v.1.0; Quality assessment of the protein model: MOLPROBITY; Size exclusion chromatography data: Chromeleon 7.2.1; Mass photometry data: DiscoverMP v2.3; Gas chromatography data: OpenLab CDS and TotalChrom v.6.3.4; Phylogenetic analysis: MAFFT, trimAI, UFBoot2, IQ-TREE2. Electronic spectrum data: Global Gauss Fit v.7. DNA sequencing: SnapGene 8.0. |

For manuscripts utilizing custom algorithms or software that are central to the research but not yet described in published literature, software must be made available to editors and reviewers. We strongly encourage code deposition in a community repository (e.g. GitHub). See the Nature Portfolio guidelines for submitting code & software for further information.

# Data

Policy information about availability of data

All manuscripts must include a data availability statement. This statement should provide the following information, where applicable:

- Accession codes, unique identifiers, or web links for publicly available datasets
- A description of any restrictions on data availability
- For clinical datasets or third party data, please ensure that the statement adheres to our policy

The data that support this study are available from the corresponding author upon request. The cryo-EM maps reported on this article are available in the Electron Microscopy Data Bank (EMDB) with the accession codes EMD-51767 (MCR activation complex + A2 after incubation with ATP) [https://www.ebi.ac.uk/emdb/EMD-51767], EMD-19787 (MCR activation complex as isolated + A2) [https://www.ebi.ac.uk/emdb/EMD-19787] and EMD-19788 (MCR activation complex as isolated without A2) [https://www.ebi.ac.uk/emdb/EMD-19788]. The local refined mask maps for A2, FeS containing subunits and Mmp3 were submitted along with EMD-51767 and EMD-19787. The atomic models are deposited in the Protein Data Bank (PDB) under the codes 9H1L (MCR activation complex + A2 after incubation with ATP) [https://doi.org/10.2210/pdb9h1l/pdb], 8S7V (MCR activation complex as isolated + A2) [https://doi.org/10.2210/pdb8s7v/pdb] and 8S7X (MCR activation complex as isolated without A2) [https://doi.org/10.2210/pdb8s7x/pdb]. Structural data used for comparison are available in the Protein Data Bank under the codes 3M32 (MCR from M. marburgensis bound to CoMS-SCoB) [https://doi.org/10.2210/pdb3m32/pdb], 1HBM (MCR from M. thermautotrophicus bound to CoMS-SCoB) [https://doi.org/10.2210/pdb1hbm/pdb], 8CRS (cryoEM structure of Mo-nitrogenase from A. vinelandii) [https://doi.org/10.2210/pdb8crs/pdb], 8DPN (cryoEM structure of Mo-nitrogenase from A. vinelandii DJ) [https://doi.org/10.2210/pdb8dpn/pdb], 6FEA (V-nitrogenase from A. vinelandii) [https://doi.org/10.2210/pdb6fea/pdb], 8OIE (cryoEM structure of Fe-nitrogenase from R. capsulatus) [https://doi.org/10.2210/pdb8oie/pdb], 8BOQ (Fe-nitrogenase from A. vinelandii) [https://doi.org/10.2210/pdb8boq/pdb], 3U7Q (Mo-nitrogenase from A. vinelandii at atomic resolution), [https://doi.org/10.2210/pdb3u7q/pdb], 3PDI (precursor-bound NifEN from A. vinelandii DJ) [https://doi.org/10.2210/pdb3pdi/pdb], and 7BI7 (P-cluster-bound NifB from M. thermoautotrophicus). Sequences for comparison with the A2 component are available in Uniprot with the accession numbers A0A8T3W5C7, D9PXN7, D7DTW8, Q7LYR5, Q58639, Q8TIZ1, A0A0E3R7T8, A0A0E3N9D1, F8ANN2, A6USB5, A0A8J8F9M9, and A0A8S9VY72. Activity assays measurements, size exclusion chromatography and spectroscopic data have been provided as Source Data. Original sequence alignments and phylogenetic trees have been deposited into Edmond, the Open Research Data Repository of the Max Planck Society, for public access and available under https://doi.org/10.17617/3.UZSFCW. All data needed to evaluate the conclusions in the paper are present in the paper and/or the supplementary information.

# Research involving human participants, their data, or biological material

Policy information about studies with human participants or human data. See also policy information about sex, gender (identity/presentation), and sexual orientation and race, ethnicity and racism.

| | |
|---|---|
| Reporting on sex and gender | N/A |
| Reporting on race, ethnicity, or other socially relevant groupings | N/A |
| Population characteristics | N/A |
| Recruitment | N/A |
| Ethics oversight | N/A |

Note that full information on the approval of the study protocol must also be provided in the manuscript.

# Field-specific reporting

Please select the one below that is the best fit for your research. If you are not sure, read the appropriate sections before making your selection.

☒ Life sciences   ☐ Behavioural & social sciences   ☐ Ecological, evolutionary & environmental sciences

For a reference copy of the document with all sections, see nature.com/documents/nr-reporting-summary-flat.pdf

# Life sciences study design

All studies must disclose on these points even when the disclosure is negative.

| | |
|---|---|
| Sample size | Sample size was not predetermined by statistical methods but three biological sample sizes and three technical replicates were used for in vitro analysis to provide datasets that enable statistics for the relevant experiments. The in vitro assays are tried-and-tested procedures that have been successfully performed by the authors in their laboratories on a day-to-day basis. The sample sizes were diligently chosen and accurately illustrate the differences throughout the various conditions. For cryoEM, two datasets containing 30,000 micrographs (incubated with ATP and 25º tilted) and 17,548 micrographs (sample as isolated, 20° tilted and non-tilted combined) were used in order to have a sufficient number of MCR's activation complex particles - with and without the A2 component - resulting in 3D reconstructions with a resolution of 2.1 (+A2 component, incubated with ATP), 2.56 (+A2 component, as isolated) and 2.78 Angstrom (-A2 component, as isolated). |
| Data exclusions | For strictly anaerobic enzymes, oxygen contamination cannot always be excluded and leads to a significant effect the experiments. If a measurement in the activity assays was obviously differing from the other replicates, this measurement was excluded from the dataset. For |

| | |
|---|---|
| | cryoEM, particles visibly containing the activation complex and/or the A2 component were chosen for determination of the high resolution structures. |
| Replication | Enzymatic assays were performed from three independent biological replicates with at least three technical repetitions each. Spectroscopic data was obtained from three independent biological replicates. Sample separation via polyacrylamide gels and size exclusion chromatography was performed at least three times from independent biological repetitions, whereas mass photometry took place from two independent biological repetitions. All replication attempts were successful. For cryoEM, one dataset of 30,000 micrographs (incubation with ATP) and a second one of 17,548 micrographs (as isolated) were analyzed. |
| Randomization | Resolution determination was carried out with two independent half sets randomly selected by the cryoSPARC processing software according to the Gold-Standard FSC procedure. For all other experiments, randomization was not applicable, since in microbiological assays all parameters are tightly controlled and therefore covariates are not relevant to our study. |
| Blinding | Blinding was not relevant to our study since no higher order species were used. Moreover we have repeated experiments from three independent biological replicates with at least three technical replicates. |

# Reporting for specific materials, systems and methods

We require information from authors about some types of materials, experimental systems and methods used in many studies. Here, indicate whether each material, system or method listed is relevant to your study. If you are not sure if a list item applies to your research, read the appropriate section before selecting a response.

## Materials & experimental systems

| n/a | Involved in the study |
|---|---|
| ☐ | ☒ Antibodies |
| ☒ | ☐ Eukaryotic cell lines |
| ☒ | ☐ Palaeontology and archaeology |
| ☒ | ☐ Animals and other organisms |
| ☒ | ☐ Clinical data |
| ☒ | ☐ Dual use research of concern |
| ☒ | ☐ Plants |

## Methods

| n/a | Involved in the study |
|---|---|
| ☒ | ☐ ChIP-seq |
| ☒ | ☐ Flow cytometry |
| ☒ | ☐ MRI-based neuroimaging |

## Antibodies

| | |
|---|---|
| Antibodies used | StrepMAB-Classic HRP (1.5 mg/mL) IBA Lifesciences, Germany/2-1509-001<br>Conjugated monoclonal antibody to detect Strep-tag®II and Twin-Strep-tag® fusion proteins |
| Validation | StrepMAB-Classic HRP (1.5 mg/mL, IBA Lifesciences, Germany/2-1509-00 counts with a Bioz Stars standard score of 93/100, supported by multiple citations in journals with an impact factor equal or higher than 10 that have reported its use in western-blot experiments. This is according to the information from https://www.iba-lifesciences.com/strepmab-classic-hrp/2-1509-001 |

## Plants

| | |
|---|---|
| Seed stocks | Report on the source of all seed stocks or other plant material used. If applicable, state the seed stock centre and catalogue number. If plant specimens were collected from the field, describe the collection location, date and sampling procedures. |
| Novel plant genotypes | Describe the methods by which all novel plant genotypes were produced. This includes those generated by transgenic approaches, gene editing, chemical/radiation-based mutagenesis and hybridization. For transgenic lines, describe the transformation method, the number of independent lines analyzed and the generation upon which experiments were performed. For gene-edited lines, describe the editor used, the endogenous sequence targeted for editing, the targeting guide RNA sequence (if applicable) and how the editor was applied. |
| Authentication | Describe any authentication procedures for each seed stock used or novel genotype generated. Describe any experiments used to assess the effect of a mutation and, where applicable, how potential secondary effects (e.g. second site T-DNA insertions, mosiacism, off-target gene editing) were examined. |

