## [Peer Review File · Nature]

Structure of the ATP driven Methyl-coenzyme M reductase activation complex

Corresponding Author: Dr Jan Schuller

Version 0:

Reviewer comments:

Referee #1

(Remarks to the Author)

The manuscript by Ramirez-Amador et al. describes the structure and activity of a putative ATP-dependent methyl-coenzyme M reductase (MCR) activation complex from *Methanococcus maripaludis*. MCR is the enzyme responsible for the production of nearly all biologically generated methane. To carry out this activity, MCR utilizes the nickel-containing coenzyme F430, which must be reduced to the Ni(I) state to be active. The authors purify and characterize a complex proposed to carry out this reductive activation, which is comprised of component A2, several methanogenesis marker proteins (Mmp3, Mmp7, and Mmp17), and a protein with a domain of unknown function (DUF2098), bound to MCR. The complex is able to generate methane in the presence of the reductant titanium(III) citrate, and this activity appears to be stimulated by the addition of ATP. Most interestingly, the structure of the complex (which was solved using cryo-EM) is asymmetrically bound to MCR with an electron transfer path leading to coenzyme F430 that consists of what appears to be three [8Fe-9S-C] clusters that resembles the precursor of the iron-molybdenum cofactor (FeMo-co) of nitrogenase, thus strengthening the evolutionary link between methanogenesis and nitrogen fixation. Together these results are of potential interest to the broad readership of Nature. However, there are a couple significant issues summarized below that need to be addressed should the manuscript be accepted for publication.

1. The demonstration of the ATP-dependent reductive MCR activation activity of the purified complex is not convincing. While there appears to be a statistically significant difference in the rates of methane formation in the presence and absence of ATP, the observed activity is only about a thousandth of a percent of the expected specific activity of an equivalent amount of purified MCRred1. Moreover, the addition of ATP only stimulates methane formation activity by ~5-fold and the distribution of the initial state(s) of MCR (i.e., MCRred1, MCRsilent, or MCRox1) in the complex is unclear. Knowing the quantitative distribution of states is important for determining whether the activation complex is functional, as titanium(III) citrate can reductively activate MCRox1 (but not MCRsilent) directly without the activation complex. From the data, there is indeed a small amount of MCRox1 present. The modest methane formation activity could therefore be due entirely to the chemical reduction of MCRox1 with titanium(III) citrate, and the observed stimulation with ATP may just be ATP-induced dissociation of the complex to generate holo MCR. Thus, it is unclear from the data if the complex facilitates the reduction of MCR to the active MCRred1 form. The proper control that would be necessary to confirm activity of the complex, wherein the activity of MCR under activation conditions without the activation complex (and with an identical distribution of states) is examined, is also missing; as currently devised, removing the activation complex from the control assay also removes MCR.
2. While compelling, it is unclear that the structure(s) of the iron-sulfur centers have been conclusively assigned. The resolutions of the cryo-EM structures of the activation complex (2.6 Å for the structure with A2 and 2.8 Å for the structure without) is quite low. The identity of the interstitial carbide of FeMo-co was not confirmed until an ultra-high 1.0 Å resolution X-ray crystal structure was obtained and combined with ¹³C-labelling and electron spin echo envelope modulation (ESSEM) spectroscopic analysis. Another approach utilized X-ray emission spectroscopy (XES) to confirm the structure of FeMo-co. Similar strategies could be utilized to confirm that the three iron-sulfur centers are indeed L-cluster-like [8Fe-9S-C] clusters. This is important to confirm the structural model and further establish the evolutionary connection between methanogenesis and nitrogen fixation. However, it should be noted that this connection has already been established by the homology of nitrogenase to the coenzyme F430 biosynthesis enzyme complex CfbCD. Thus, care should be taken to make clear in the text that this is not a novel concept, but rather the data better illuminates the extent to which these pathways are intertwined.

Referee #2

(Remarks to the Author)

My expertise is in molecular evolution, I will thus submit my evaluation of the phylogenetics based evolutionary claims declared in the manuscript. Overall, the general evolutionary and phylogeny framing of the paper (and I believe one of the main reasons behind the claimed novelty) has several flaws.

To start, the connection between methanogenesis biosynthesis proteins and nitrogenase is not new (e.g., see Boyd et al), and it's also known that *nifB* is needed to support methanogenesis. The claim that the earliest role of *nifB* was for methanogenesis (instead of N-fixation) is new, however there are major issues with their tree reconstructions and inferences thus this new claim lacks supporting evidence. At a top level, unfortunately, the tree reconstruction seems sloppy and needs more care; a well-rooted *nifB* tree could provide novel evidence here, but I can't understand their description of the rooting method and their phylogeny-based claims seems possibly circular. Based on the figures (they are a little unclear and hard to read, I did my best) methanogens are placed in the deepest branching lineages, so naturally one would infer an early methanogen-associated role for *nifB*.

Similarly, the rest of this section relating to CbfD is also not new (see again Boyd et al. *Geobio* 2011), and there isn't any phylogenetic evidence presented in the paper specifically to support the rooting for that either. In general, they would have to make some other independent and convincing rationale for the rooting. The conclusions are bold and flashy but naturally one looks at the data (i.e., the tree) as they currently stand aren't the best evidence.

Further details:

"A different root would imply more horizontal gene transfers between archaea and, consequently, becomes less likely"
What is the evidence for this?

"After mapping the presence of the nitrogenase catalytic subunits NifK and NifD29-31 onto our tree, the data suggests that nitrogenase was absent at the root node of the NifB phylogeny. This supports the idea that NifB's role during MCR activation precedes its association with nitrogenases."

It's not obvious to me looking at Extended Data Fig. 15 that nitrogenases were absent in organisms with oldest ancestral NifB, rather than secondarily lost in some archaeal lineages. Further, it's unclear from the methods whether amplicons/incomplete genome assemblies were included as sources of NifB sequences. Therefore, the missing nitrogenase genes is likely an artifact of the sequence search strategy. Lastly, make sure your represented taxa is available and easy to access, it is extremely hard to replicate/understand their analyses without this.

"Inferring from our NifB phylogeny that methanogenesis originated earlier than nitrogen fixation..."
Because the rooting method for the NifB tree is unclear (see first comment), I'm not convinced that this conclusion is specifically supported by the NifB tree.

"...we then rooted this tree with CbfD. These results imply that nitrogenases have evolved from enzymes able to convert porphyrin-based substrates"
It has previously been established that CbfD is homologous to NifDKEN, and it's also previously been proposed that a CbfD-like protein was ancestral to NifDKEN (e.g., see Boyd et al., *Geobiology*, 2011). It's therefore misleading to claim that this is a new finding. In addition, from the preceding comment, it is not appropriate to root the nitrogenase tree with CbfD. An independent rationale for the rooting is needed to support their flashy claims.

"Together, our results strongly suggest that the M-clusters of nitrogenases originate from the methanogenic FeS clusters in the MCR activation complex."

Metalloclusters don't have "ancestry", the genes that make them do.

Referee #3

(Remarks to the Author)

The manuscript by Ramirez-Amador et al of the ATP-dependent methyl-coenzyme M reductase activation complex reveals the structure of the complex responsible for the reductive activation of F430 to the Ni(I) state. Three FeS clusters are present in the activation complex that astonishingly resemble the nitrogenase FeMo-cofactor; these clusters form an electron transfer pathway towards F430. This is the first example of this type of metallocluster outside of nitrogenase and is an extraordinary observation with potentially far-reaching implications for the evolution of the biosynthetic pathway for this cluster. Present in the activation complex is the ATP-binding cassette (ABC) fold containing A2 subunit that likely is involved in rearrangements powered by ATP binding and hydrolysis. Overall, this is remarkable study and I enthusiastically support publication.

A few comments for the authors consideration

line 112 - how does this activity (16 nmol CH₄/120 min/10 ug) compare to that reported for the specific activity of MCR?

line 131 - "one of the MCR heterodimers" - should this be heterotrimer?

line 143 - another example of an enzyme with an ABC subunit is the carbon-phosphorus lyase reported by Amstrup et al *Nat. Comm.* 14, 1001 (2023). In this system, "ATP hydrolysis induces drastic structural remodeling leading to opening of the core complex and reconfiguration of a metal-binding and putative active site", not unlikely what is proposed here.

line 188 - as near as I can tell from Figure 4 and extended Figure 13, the densities for FeSI, II and III clearly resemble that for the nitrogenase FeMo-cofactor. However, without looking at the actual maps, it is not possible to make this assessment with full confidence.

line 250 - has the consequence of a NifB deletion on the MCR activation machinery been studied? Since the native complex is tagged with Twin-Strep tag using the CRISPR-Cas12a system, this should be possible.

line 282 - the structure of the ADP-AIF stabilized nitrogenase complex provided the first demonstration of how nitrogenase operates as a nucleotide dependent switch (Schindelin et al Nature 387, 370 (1997)). While both nitrogenase and ABC proteins couple nucleotide binding and hydrolysis to protein conformational changes, they have distinct protein folds. The archerases discovered by W. Buckel (see FEMS Microbiol. Rev. 28, 455 (2004)) should also be included in this list.

As a general comment for speculation - why are 3 copies of such an elaborate metallocluster be needed to activate F430 - are they "only" serving as electron carriers or might they have more complicated roles?

Version 1:

Reviewer comments:

Referee #1

(Remarks to the Author)

In the previous version of the manuscript by Ramirez-Amador et al. there were concerns raised regarding the ATP-dependent activity assays of the activation complex, specifically regarding the low specific activity of the activated MCR, the small observed rate stimulation by ATP, the presence of MCRox1 in the sample (which can be activated by the reductant directly), and the lack of a control in the absence of the activation complex. These concerns were not completely addressed. However, the addition of a control where alkaline phosphatase was included in the reaction mixture (as well as mass photometry experiments) provides strong support that the reductive activation was indeed ATP dependent and the activity derived from the direct activation of MCRox1 was negligible.

Concerns were also raised regarding the definitive assignment of the structures of the complex iron sulfur clusters, i.e., whether there was a central carbide similar to the structure of FeMoco in nitrogenase. While the suggested, more definitive methods of confirmation were not pursued, additional local refinement of the structure and the inclusion of a figure showing the density and distances of the atoms in the clusters and a table comparing these with FeMoco, FeFeco, and the L- and K-clusters help to make the argument more compelling (Extended Data Table 5). Together these additions have improved the manuscript such that it is suitable for publication in Nature. Below are some minor items to address prior to publication.

It is stated in the Methods section and rebuttal that resolutions up to 1.8 Å were obtained in the newly refined data but the in the Abstract it still says 2.1 Å.

Line 52: The structure of the macrocycle of the F430 is technically not a porphyrin.

Line 55: F430 can exist in +1, +2, and +3 oxidation states but does not transition between all three during the catalytic cycle.

Line 80: Activation relies on ATP hydrolysis (not relays).

Line 110: The Ni(III)-MCRox1 state not only is not ruled out by the UV-vis data, its presence in the activation complex sample was confirmed via EPR (Fig. 4d). Thus stating that it is below the detection limit is misleading.

Line 133: Similar to the catalytic cycle of nitrogenase, ATP binding, hydrolysis, and release is likely coupled to MCR activation complex formation, F430 reduction, and complex dissociation. The mass photometry data does suggest that ATP binding promotes complex formation, but it is somewhat confusing to say it doesn't play a role in the dissociation of the complex (and by itself does not rule out that the direct reduction of MCRox1 could be occurring).

Line 221: May be less confusing to refer to the clusters as [8Fe-9S-C] as opposed to double cubane [4Fe-4S] clusters.

Line 310: You should include the actual calculated specific activity of your enzyme preparations for comparison. If my calculations were correct, the highest activation that was observed was approximately 5-fold less than that from the A3a complex.

Fig. 4. Panel d mislabeled as f.

Referee #2

(Remarks to the Author)

I don't have any further comments.

Referee #3

(Remarks to the Author)

In this revised manuscript, the authors have satisfactorily addressed my concerns, and I continue to be enthusiastic about this work.

Referee #1 (Remarks to the Author):

The manuscript by Ramirez-Amador et al. describes the structure and activity of a putative ATP-dependent methyl-coenzyme M reductase (MCR) activation complex from *Methanococcus maripaludis*. MCR is the enzyme responsible for the production of nearly all biologically generated methane. To carry out this activity, MCR utilizes the nickel containing coenzyme F430, which must be reduced to the Ni(I) state to be active. The authors purify and characterize a complex proposed to carry out this reductive activation, which is comprised of component A2, several methanogenesis marker proteins (Mmp3, Mmp7, and Mmp17), and a protein with a domain of unknown function (DUF2098), bound to MCR. The complex is able to generate methane in the presence of the reductant titanium(III) citrate, and this activity appears to be stimulated by the addition of ATP. Most interestingly, the structure of the complex (which was solved using cryo-EM) is asymmetrically bound to MCR with an electron transfer path leading to coenzyme F430 that consists of what appears to be three [8Fe-9S-C] clusters that resembles the precursor of the iron-molybdenum cofactor (FeMo-co) of nitrogenase, thus strengthening the evolutionary link between methanogenesis and nitrogen fixation. Together these results are of potential interest to the broad readership of Nature. However, there are a couple significant issues summarized below that need to be addressed should the manuscript be accepted for publication.

1. The demonstration of the ATP-dependent reductive MCR activation activity of the purified complex is not convincing. While there appears to be a statistically significant difference in the rates of methane formation in the presence and absence of ATP, the observed activity is only about a thousandth of a percent of the expected specific activity of an equivalent amount of purified MCRred1. Moreover, the addition of ATP only stimulates methane formation activity by ~5-fold and the distribution of the initial state(s) of MCR (i.e., MCRred1, MCRsilent, or MCRox1) in the complex is unclear. Knowing the quantitative distribution of states is important for determining whether the activation complex is functional, as titanium (III) citrate can reductively activate MCRox1 (but not MCRsilent) directly without the activation complex. From the data, there is indeed a small amount of MCRox1 present. The modest methane formation activity could therefore be due entirely to the chemical reduction of MCRox1 with titanium (III) citrate, and the observed stimulation with ATP may just be ATP-induced dissociation of the complex to generate holo MCR. Thus, it is unclear from the data if the complex facilitates the reduction of MCR to the active MCRred1 form. The proper control that would be necessary to confirm activity of the complex, wherein the activity of MCR under activation conditions without the activation complex (and with an identical distribution of states) is examined, is also missing; as currently devised, removing the activation complex from the control assay also removes MCR.

We would like to extend our gratitude to the reviewer for the asserted comments, and we believe this feedback has significantly improved our manuscript.

*As the referee remarks, we recognize that our activity values are notably low when compared to other reports on the activation of MCR, which reach up to 50-100 $\mu\text{mol}/\text{min}\cdot\text{mg}$ (Goubeaud 1997, Prakash 2014). However, crucial differences exist since we are not assessing the activity of a natively purified MCR from thermophilic methanogens (i.e., *Methanothermobacter marburgensis*, *Methanothermobacter thermoautotrophicus*). Given the abundance of MCR in methanogens (ca. 10% of proteome), native purification offers high protein yields that allow for robust enzymatic assays. Instead, by following an affinity purification strategy, we report the activity of MCR isolated from the mesophile *M. maripaludis* and – as a particular milestone of this work – we characterized a transitory biogenesis intermediate of MCR bound to its activation complex. We emphasize the challenges of low protein yield, since we obtained no more than 0.5 mg of protein from 10 liters of culture, hinting at the lower abundance of McrC compared to McrABG. Additionally, even though we utilized 1.25 μM (or 12 μg) of total protein for our reactions, such samples are not composed of MCR exclusively but also other activation components summarized in the Extended Data Table 3.*

Extended Data Table 3. Protein components present in the MCR activation complex determined by MS.

Name	Description	Mol. Size (kDa)	Uniprot ID
McrA	Methyl-coenzyme M reductase subunit alpha	61.1	A0A2L1CBB0
A2	Methyl-coenzyme M reductase system component A2	59.5	A0A2L1C9A1
Mmp3	Methanogenesis marker protein 3	56.4	A0A2L1CAI0
McrB	Methyl-coenzyme M reductase subunit beta	46.7	A0A2L1CBB3
Mmp7	Methanogenesis marker protein 7	34.9	A0A2L1C9H0
McrG	Methyl-coenzyme M reductase subunit gamma	29.6	A0A2L1CBG2
McrC	Methyl-coenzyme M reductase operon protein C	21.3	A0A2L1CBQ8
Mmp17	Methanogenesis marker protein 17	21.1	A0A2L1C8U1
DUF2098	Domain of unknown function containing protein 2098	10.6	A0A2L1CAX0

Addition in the main text:

“Upon the addition of Ti(III)-citrate, earlier studies on the activation of MCR achieved a maximum activity of 0.1 $\mu\text{mol CH}_4 \text{ min}^{-1} \text{ mg}^{-1}$ (or 0.1 U/mg), not much different from our results after 2 hours of reaction. Nevertheless, such values remain low when compared to the average activity of mature MCR from *Methanothermobacter* species (20 U/mg, being 100 U/mg the maximum ever reported). On the other hand, our purification strategy proved to be successful capturing a low-abundance and highly transient biogenesis intermediate, challenging a direct comparison with a fully mature MCR. Such discrepancy also indicates an incomplete understanding of the activation process, insinuating that additional factors might still be required to efficiently reduce the nickel ion in F_{430} .”

We emphasize that our primary focus is confirming the ATP-dependent activity of our protein complex. While fully acknowledging its limitations, our findings align well with earlier reports on the activation of MCR utilizing the A3a fraction (Rouviere 1988 & 1989, Prakash 2014). However, we now refine this understanding by identifying a minimal set of essential components (McrC, Mmp7, Mmp17, Mmp3 and DUF2098) required for activation. Importantly, we observed no activity when Ti(III)-citrate ($E^\circ = -480 \text{ mV}$) was replaced with DTT at pH 7 ($E^\circ = -320 \text{ mV}$) (Extended Data Fig. 2f), contrary to previous data by Prakash 2014. We discuss these implications and recognize that our understanding of the full activation process remains incomplete, suggesting that additional factors in the A3a fraction may play a role in lowering the redox potential of F_{430} ($E^\circ = -650 \text{ mV}$ as free ligand) for DTT to reduce it under these conditions.

Addition in the main text:

“In support of this, no activity was detected when Ti(III)-citrate was replaced by DTT (Ext. Data Fig. 2f) – contrary to previous observations – suggesting that additional proteins in the A3a fraction may be critical for lowering the redox potential barrier and activate F_{430} . In *M. marburgensis*, the genes encoding for additional Mmp proteins are clustered together in a single open reading frame (MTBMA c04360 to c04410, interestingly followed by *nifB*), while *Methanosarcina* species even contain the gene for the A2 component within an equivalent locus. Thereby, homologs of these extra Mmp proteins are presumably still required in *M. maripaludis* to activate F_{430} under physiological conditions, although here they are widely dispersed around the genome. All combined, our data describe a minimal number of proteins able to bind and activate MCR.”

Extended Data Fig. 2f. All CH_4 production assays, including complementary conditions (i.e., non-hydrolyzable AMP-PNP, ATP but using **DTT** instead of Ti(III)-citrate and both **‘as isolated’** and **‘ATP’ without pre-activating conditions** (*, see Methods). All values are normalized to micrograms of total protein. [...]

Aiming to optimize our assay conditions, we introduced a ‘pre-activation’ step to mitigate any inhibitory effect caused by an early formation of CoMS-SCoB (Fig. 1c and Extended Data Fig. 2f [*]). As a result, the pre-activated enzyme produced at least twice the amount of CH_4 in comparison to the non-activated samples (Extended Data Fig. 2f) per microgram of protein. The pre-activation consisted in incubating 5 μM of enzyme at 37°C and pH 7 for 30 minutes in the presence of only $\text{CH}_3\text{-CoM}$, Ti(III)-citrate and ATP (see Methods). The rationale of this pre-activation is to increase the proportion of reduced F_{430} while preventing the heterodisulfide by-product since CoB-SH is excluded in the initial step. This approach was inspired by reported evidence (Rouviere 1988, Ellermann, 1988, Prakash 2014), and supported by our structural data showing the heterodisulfide-bound state blocking the proximal active site (Fig. 3b). Only after pre-activation, 1.25 μM (12 μg) of enzyme was mixed with the aforementioned components plus CoB-SH and cyanocobalamin. The combined function of cyanocobalamin and Ti(III)-citrate efficiently reduces the heterodisulfide back to CoM-SH and CoB-SH. Consequently, all described assays were performed with pre-activated enzyme.

Addition in the main text:

“Before the activity assay, 5 μM of enzyme were pre-incubated for 30 min at 37°C and pH 7 in the presence of Ti(III)-citrate ($E^\circ = -480 \text{ mV}$), $\text{CH}_3\text{-S-CoM}$ and ATP in order to avoid inhibitory effects by the immediate formation of CoMS-SCoB (see Methods). Then, 12 μg of activated protein were added to the mixture containing both $\text{CH}_3\text{-S-CoM}$ and CoB-SH as well as the remaining reaction components (see Methods). The reaction produced $40 \pm 6 \text{ nmol}$ of CH_4 after 2 h at 37°C and pH 7 when the sample was pre-incubated in the presence of 5 mM ATP, doubling the CH_4 produced by the

non-activated enzyme (Extended Data Fig. 2f). In the assay without the addition of ATP ('as isolated'), only 9 ± 2 nmol of CH_4 were produced (Fig. 1c)."

→Fig. 1c. CH_4 production assays showing the ATP-dependent activation of MCR *in vitro*. [...] Control: reaction with no protein; -ATP: depletion of ATP with alkaline phosphatase (see Methods); as iso: 'as isolated'; +ATP: external addition of ATP.

Fig. 3b. Proximal active site containing the CoMS-SCoB by-product. [...]←

We suggest that the basal generation of CH_4 from the 'as isolated' sample is due to co-purified ATP bound to the A2 component – as evidenced by our cryoEM model in both 'as isolated' and ATP-incubated sample (Fig. 2d, see enclosed PDB files and electron density maps). Despite our efforts, vitrification of the enzyme under turnover conditions was unsuccessful – because of the Ti(III)-citrate background.

Addition in the main text:

"The basal production of CH_4 is likely due to trace amounts of ATP co-purified with the protein complex (see below)."

[...]

"Importantly, our structure revealed two ATP molecules and their coordinating Mg ions (Fig. 2d), indicating that the structure captures the enzyme in a pre-hydrolysis state. Such finding reinforces the basal production of CH_4 in the experiment where no ATP was added *in vitro* (Fig. 1c)."

Fig. 2d. A2 component sitting on the backside of McrG. ATP molecules bind in both active sites, suggesting a pre-hydrolytic state (contour level 0.08σ). The Mg ions are shown as pink spheres.

Strikingly, when the co-purified ATP was removed from the reaction with alkaline phosphatase (see Methods), no CH_4 formation was detected. This result conclusively demonstrates that the activation of MCR in our assays is strictly ATP-dependent, and Ti(III)-citrate itself cannot reduce the nickel ion in F_{430} under these conditions (pH 7 and 37°C). When ATP in the system was replaced by the non-hydrolyzable ATP-homolog AMP-PNP, we observed a slight increase in the formation of CH_4 vs the sample 'as isolated'. This hints already at a positive contribution by nucleotide-binding into the activation of F_{430} (see below).

Extended Data Fig. 2f. All CH_4 production assays, including complementary conditions (i.e., non-hydrolyzable AMP-PNP, ATP but using DTT instead of Ti(III)-citrate and both 'as isolated' and 'ATP' without pre-activating conditions (*, see Methods). All values are normalized to micrograms of total protein. [...] Control: reaction with no protein; **-ATP: depletion of ATP with alkaline phosphatase (see Methods)**; as iso: 'as isolated'; +ATP: external addition of ATP.

To provide further insights in the role of ATP-binding in the molecular architecture of the complex, we performed mass photometry analysis (Extended Data Fig. 2g). Interestingly, the mass photometry results (with 0.03 μM of protein) clearly resemble observations from our cryoEM datasets (Fig. 1d, Extended Data Figs. 3 and 4). Here, the ‘as isolated’ sample showed a peak matching the size of MCR + activation complex accompanied by a series of smaller subunits in a free state. Among these, a peak around 63 kDa closely matches with the molecular weight of the A2 component (59.5 kDa). This is in line with our initial cryoEM data observations, where only 58% of particles contain the A2 component, whereas the remaining particles lack this subunit (Extended Data Fig. 3c). As a general comment, no free MCR was found in any of the cryoEM datasets (2.6 μM protein utilized for vitrification).

When the protein was incubated with ATP, the mass photometry still showed a peak for MCR + activation complex but the smaller populations were now condensed into one defined peak (ca. 100 kDa – likely a combination of Mmps under the dilution effects since 0.03 μM of protein were utilized). On top of this, cryoEM data processing of the sample +ATP strikingly resulted in 92% of particles containing the A2 component (Extended Data Fig. 4c). Altogether, our comprehensive mass photometry analysis and the newly resolved high-resolution cryo-EM structure (up to 1.8 \AA) following ATP addition, demonstrate that ATP enhances the binding of the A2 component to the larger activation complex. Thus, our data suggest that ATP does not play a role in the disassembly of the activation complex to generate free MCR, but rather facilitates its interactions with the A2 component. Moreover, the use of non-hydrolyzable AMP-PNP in mass photometry (Extended Data Fig. 2g) almost equals the profile obtained when ATP was added, confirming that nucleotide-binding indeed favors the coupling of A2 with MCR.

Addition in the main text:

“We further analyzed the effect of ATP on the formation of the activation complex employing mass photometry (MP) as a single-molecule level technique (Extended Data Fig. 2g). For the sample ‘as isolated’, MP confirmed an assembly of 502 kDa and other defined peaks of smaller size, indicating partial disruption of the complex when extensively diluted (0.03 μM protein). Importantly, complementing the sample with ATP stabilized the binding of smaller subunits onto the larger MCR activation complex. A similar behaviour is observed when AMP-PNP is utilized instead of ATP (see Discussions). This demonstrates that ATP-binding promotes the assembly of the A2 component with MCR, rather than dissociation of the activation complex, suggesting that the function of the A2 component is crucial for a full-complex formation.”

Extended Data Fig. 2g. Mass photometry histograms of MCR+activation complex as isolated (left), and after incubation with 20 mM Ti(III)-citrate plus 5 mM ATP (middle) or 5 mM AMP-PNP (right).

Finally, we employed HPLC-coupled UV-vis absorption spectroscopy to determine the oxidation states of Ni-F₄₃₀ in our preparations. We took advantage of the photodiode detector in our SEC system to directly inspect the UV-vis spectra of protein complexes differing in their molecular size. Here, independent spectra were recorded for the activation complex-bound MCR and the free MCR core (Fig. 1b and Ext. Data Fig 2c,d,e). We presume the latter as an effect of extensive dilution in the system – agreeing with a ca. 275 kDa shoulder observed by mass photometry. Several reports (Goubeaud 1997, Becker & Ragsdale 1998, Duin 2011, Ragsdale 2017) describe the UV-vis features of Ni-F₄₃₀ in MCR according to the following consensus: Ni(I) - maximum absorbance at 376-388 nm and 750 nm; Ni(II) - maximum absorbance at 420 nm and a shoulder at 440-445 nm; and Ni(III) - maximum absorbance 420-425 nm and a shoulder at 390 nm. The MCR core spectra has a maximum absorbance at 420 nm with a smaller shoulder at 447 nm, corresponding to a Ni(II)-MCR_{silent} state. MCR + activation complex shares features of Ni(II)-MCR_{silent}, although additional absorbance at 412 nm (likely from iron-sulfur metallocofactors) and the background at lower wavelengths cannot completely discard Ni(III)-MCR_{ox1}. Since cells were not “treated” in any particular way to induce the MCR_{ox1} state (Rospert 1991, Goubeaud 1997, Becker & Ragsdale 1998, Zhou 2013) a majority of inactive Ni(II)-MCR is expected, but the MCR_{ox1} spectra observed in our EPR experiment (ca. 10 μM protein) suggests a portion of MCR resting in the Ni(III) state (Fig. 4d and Extended Data Fig. 8j). Importantly, we remark that even MCR activation reports achieving significant amounts of the active Ni(I)-MCR_{red1} state, still show significant contributions from Ni(III)-MCR_{ox1} (Goubeaud 1997, Becker & Ragsdale 1998). Incubating the sample with Ti(III)-citrate + ATP before injection in the SEC system did not change the UV-vis features shown in the Extended Data Fig. 2c.

Addition in the main text:

“UV-vis spectra – extracted from each elution peak in the chromatogram – showed that both the activation complex-bound MCR and the MCR core share a maximum absorbance around 420 nm with a visible shoulder at 447 nm, which is more pronounced in the MCR core (Extended Data Fig. 2d,e). These electronic features are consistent with those reported for Ni(II)-MCR_{silent}, leading us to conclude that most of the F₄₃₀ rests in the Ni(II) state in both the biogenesis intermediate and MCR. Ni(III)-MCR_{ox1} features in the UV-vis spectrum cannot be discarded but may lie below the detection limit at the measured concentration.”

[...]

“The low-field signal ($g_1 = 2.23$, $g_2 = 2.17$, $g_3 = 2.15$) is most prominent at higher temperatures and can be assigned to the MCR_{ox1} state, accepted to form a Ni(III)-thiolate potentially in equilibrium with a high spin Ni(II) coupled to a thiyl-radical form. According to previous studies, the EPR signal of MCR_{ox1} might partially persist strong reducing conditions and coexist with Ni(I) or Ni(II) species. Thus, we observe minor amounts of MCR_{ox1} in our sample, based on the EPR data. However, as evidenced by the UV-vis bands (Extended Data Fig. 2d,e), a major part of the F₄₃₀ in our sample seems to rest in an EPR-inactive Ni(II)-MCR_{silent} state.”

→Fig. 1c. Separation of MCR activation complex (500 kDa) and the MCR core (298 kDa) by SEC. [...]

Extended Data Fig. 2d. UV-vis absorption spectra of MCR + activation complex (left) and MCR core (right). Dashed lines denote 386 nm, 412 (red), 420 and 447 nm. [...]

We thank the referee for the observation on the capacity of Ti(III)-citrate to directly activate F₄₃₀. This information is indeed valuable for the reader and we appreciate the referee for bringing this to our attention.

Since our activity assays are conducted at pH 7, it is unlikely that the midpoint redox potential of Ti(III)-citrate (-480 mV) accounts for the activation of MCR_{ox1} present in our sample (Zehnder 1976, Holliger 1993). Instead, the reduction of MCR_{ox1} by Ti(III)-citrate has been rigorously shown under alkaline conditions (pH 9-10; Goubeaud 1997, Becker & Ragsdale 1998). At 25°C the midpoint redox potential of Ti(III)-citrate is reduced approximately -60 mV per pH unit in the range between 7-10 (Zehnder 1976), making the reduction of nickel possible in F₄₃₀. Prakash et al. (2014) demonstrated the ATP-induced activation of MCR at pH 7 with DTT (-320 mV) but only when the whole A3a fraction is added into the mixture. As we highlighted in our previous comments, other external components (i.e., uncharacterized Mmps), combined with the role of ATP, likely modulate the redox potential of F₄₃₀ for the activation to occur. Another breakthrough from these authors was the fact that both MCR_{ox1} and MCR_{silent} can be activated. Such information agrees with our observations on the redox states of F₄₃₀ as well as our activity assays. As detailed above, most of F₄₃₀ seems to rest in the EPR-silent inactive Ni(II)-state while co-existing with a certain proportion of Ni(III)-MCR_{ox1}. The latter has been explained as a consequence of CoMS-SCoB inhibition (Prakash et al. 2014, Thauer 2019), in correspondence to the ligand asymmetry we found within the active sites of MCR (Fig. 3).

Additions in the main text:

“From this perspective, the binding of ATP in A2 facilitates the interactions with MCR as well as the activation of F₄₃₀ – confirmed by the production of CH₄. Remarkably, when ATP is replaced by the non-hydrolyzable AMP-PNP, not only the stabilization effect is maintained but also a moderate increase in the CH₄ levels is observed (Extended Data Fig. 2f,g). This means that conformational states influenced by nucleotide-binding (i.e., the lost coordination of Q151^{McrA}) already affect activation and potentially modulate the redox potential of F₄₃₀. As a free ligand, F₄₃₀ has a midpoint redox potential of -650 mV and only Ti(III)-citrate at pH 9-10 had shown enough reducing power for its activation (Goubeaud 1997, Becker 1998). However, the binding and hydrolysis of ATP could theoretically increase the redox potential in a way that Ti(III)-citrate ($E^{\circ} = -480$ mV at pH 7) is able to reduce Ni(II) to Ni(I). In a similar fashion, during the ATP-dependent activation of some corrinoid cofactors, ATP hydrolysis drastically increases the midpoint redox potential of the Co(II)/Co(I) couple (-500 to -250 mV). Moreover, additional Mmp interactors may push the redox potential of F₄₃₀ even further, potentially allowing the reduction of nickel by milder reducing agents like DTT ($E^{\circ} = -320$ mV) and physiologically relevant electron donors such as reduced ferredoxin ($E^{\circ} = -420$ mV). Altogether, this places the A2 component’s function on the activation of MCR alongside other prominent nucleotide-switching machines able to drive catalytic activation or product release. Although further details on the redox cycle of F₄₃₀ during its activation are still unknown, both ATP binding and hydrolysis seem to trigger such complex mechanism”.

We would like to highlight our main findings on the role of ATP in the activation of MCR:

- I. In the absence of ATP, neither Ni(II)-MCR_{silent} nor MCR_{ox1} are activated by Ti(III)-citrate alone under the tested conditions (37°C and pH 7).
- II. The addition of ATP does not induce the disassembly of the complex to release free MCR but rather promotes interactions with the A2 component.
- III. ATP hydrolysis and – to a smaller extent – ATP binding seem to modulate the redox potential of F₄₃₀ to facilitate the reduction of nickel. Conformational changes in MCR while binding the activation complex, as well as the role of additional proteins (such as those present in A3a), may contribute to this as well.

References:

- Goubeaud et al (1997), <https://febs.onlinelibrary.wiley.com/doi/10.1111/j.1432-1033.1997.00110.x>
- Prakash et al (2014), <https://journals.asm.org/doi/full/10.1128/jb.01658-14>
- Rouviere et al (1988), <https://journals.asm.org/doi/10.1128/jb.170.9.3946-3952.1988>
- Rouviere et al (1989), <https://journals.asm.org/doi/10.1128/jb.171.9.4556-4562.1989>
- Ellermann et al (1988), <https://febs.onlinelibrary.wiley.com/doi/10.1111/j.1432-1033.1988.tb13941.x>
- Becker and Ragsdale (1998), <https://pubs.acs.org/doi/10.1021/bi972145x>
- Duin et al (2011), <https://www.sciencedirect.com/science/article/pii/B9780123851123000093>
- Ragsdale et al. (2017), <https://doi.org/10.1039/9781788010580-00149>
- Rospert et al (1991), [https://doi.org/10.1016/0014-5793\(91\)81323-Z](https://doi.org/10.1016/0014-5793(91)81323-Z)
- Zhou et al (2013), <https://www.frontiersin.org/journals/microbiology/articles/10.3389/fmicb.2013.00069/full>
- Zehnder (1976), <https://www.science.org/doi/10.1126/science.793008>
- Holliger et al (1993), <https://pubs.acs.org/doi/10.1021/ja00066a034>
- Thauer et al 2019, <https://pubs.acs.org/doi/10.1021/acs.biochem.9b00164>

2. While compelling, it is unclear that the structure(s) of the iron-sulfur centers have been conclusively assigned. The resolutions of the cryo-EM structures of the activation complex (2.6 Å for the structure with A2 and 2.8 Å for the structure without) is quite low. The identity of the interstitial carbide of FeMo-co was not confirmed until an ultra-high 1.0 Å resolution X-ray crystal structure was obtained and combined with ¹³C-labelling and electron spin echo envelope modulation (ESEEM) spectroscopic analysis. Another approach utilized X-ray emission spectroscopy (XES) to confirm the structure of FeMo-co. Similar strategies could be utilized to confirm that the three iron-sulfur centers are indeed L-cluster-like [8Fe-9S-C] clusters. This is important to confirm the structural model and further establish the evolutionary connection between methanogenesis and nitrogen fixation. However, it should be noted that this connection has already been established by the homology of nitrogenase to the coenzyme F₄₃₀ biosynthesis enzyme complex CfbCD. Thus, care should be taken to make clear in the text that this is not a novel concept, but rather the data better illuminates the extent to which these pathways are intertwined.

To provide a more precise characterization of the iron-sulfur centers, we gathered our best efforts in cryoEM single-particle analysis to determine a high-resolution structure of the MCR's activation complex pre-incubated with ATP (as we found that ATP stabilized the A2 component onto MCR). Here, we collected data by pre-tilting the microscope's stage up to 25° to overcome preferred orientation effects. Strikingly, even though pre-tilting may have an impact on protein's resolution, we reached a global resolution of 2.1 Å with local resolutions as high as 1.8 Å at the active pockets and cluster binding sites. Importantly, only 0.7% of EMDB entries have resolutions below 2 Å, and 17.2% fall within the 2-3 Å range – underscoring the significance of our structural accomplishment (https://www.ebi.ac.uk/emdb/statistics/emdb_resolution_distribution). From this improved map, we could not only state that the electron densities of the clusters resemble those of the [8Fe-9S-C] (L-clusters) but even accurately fit the center of individual sulfur atoms. We determined the bond distances of Fe-Fe (avg. 2.61 Å), Fe-S (avg. 2.25 Å) and Fe-C (avg. 1.99 Å) (Fig. 4c) to closely align in geometry with those in L- and M-clusters of nitrogenase (such as FeFeco and FeMoco). Nevertheless, these metrics differ from those of other "bridged" clusters like the K-cluster (an earlier precursor of M-clusters). A detailed comparison has been provided in Extended Data Table 5.

Additional evidence to confirm the interstitial carbide ion is desirable to fully recognize these cofactors as L-clusters in our cryoEM density. For this reason, we have strengthened our argument by highlighting the geometric and topological similarities between the clusters found in the MCR activation complex and nitrogenase metallocofactors. However, we completely agree with the referee's suggestion of employing powerful spectroscopic techniques such as X-ray emission spectroscopy (XES, Lancaster et al 2013), electron-spin echo envelope modulation (ESEEM, Spatzal et al 2011) and radiolabeling experiments (Pérez et al 2021). These advanced methods would unambiguously serve as validation of the interstitial carbide ion at the center of the L-clusters. At present, the transient activation assembly of MCR which we have been capturing exists only as a very small fraction (<0.5 mg protein from 10 L cells) compared to the MCR core alone. Therefore, currently it is not feasible to validate the presence of the carbide ion with the proposed spectroscopic methods, which require larger amounts of protein sample. However, we have now expanded the discussion section to clarify the necessity of direct evidence on such central carbide ion.

Addition in main text:

“Strikingly, three uniquely coordinated FeS clusters (Fig. 4) constitute an electron transfer pathway for the activation of F_{430} . Whereas the high-resolution electron density of such cofactors clearly suggests the [8Fe-9S-C] topology of an L-cluster (Extended Data Fig. 7 and Extended Data Table 5), the electron weak carbide ion in the center – distinguished as a hole – requires thorough spectroscopic efforts for full confirmation. Techniques such as valence-to-core X-ray emission spectroscopy (V2C XES), electron spin echo envelope modulation (ESSEM) and C^{13} -labeling coupled to ENDOR spectroscopy have all successfully identified the interstitial carbide signal in both M- and L-clusters. Similar methods could be employed to fully characterize the metallocofactors binding within the activation complex of MCR.”

Fig. 4c. Electron density cloud of sulfur atoms in FeSII at a contour level of 0.32σ . Bond lengths are shown as dashed lines and color depends on the atom pair (Fe-Fe in orange, Fe-S in black and Fe-C in gray). [...]

Extended Data Table 5. Bond metrics comparison between the FeS clusters detailed in this study and similar topologies.

Bond / Å	FeS ^a	FeFeco (8BOQ)	FeMoco (3U7Q)	Fe ₈ S ₈ (3PDI)	K-cluster (7BI7)
Fe-C	1.99 ± 0.01	2.00 ± 0.01	1.99 ± 0.01	NA ^{b,e}	NA ^b
Fe-S	2.25 ± 0.03	2.26 ± 0.04	2.25 ± 0.03 (2.36 ± 0.01) ^c	2.28 ± 0.02	2.36 ± 0.1
Fe-Fe	2.61 ± 0.03	2.62 ± 0.03 (2.82 ± 0.09) ^d	2.64 ± 0.03 (2.69 ± 0.029) ^d	2.65 ± 0.01	2.69 ± 0.11

^a this study

^b not applicable.

^c Mo-S.

^d Fe8-Fe.

^e published before demonstrating the presence of a central carbide ion.

Regarding the discussion on CfbD as a known link between both methanogenesis and the origin of nitrogenases, we have now taken care of our writing to establish this was an already accepted and largely implied concept. We have expanded both the results and discussion sections to a more thorough and detailed analysis.

Addition in main text:

“As suggested by our phylogenetic reconstructions, these cofactors were then incorporated into a protein scaffold most likely deriving from the F_{430} biogenesis enzyme CfbD – whose relationship to nitrogenases has been known for some time. Thus, such protein scaffold and the catalytic cofactors of the nitrogenase family (i.e., FeMoco) would ultimately derive from the same physiological function: the biogenesis and activation of coenzyme F_{430} .”

References:

Lancaster et al (2013), <https://pubs.acs.org/doi/10.1021/ja309254g>

Spatzal et al (2011), <https://www.science.org/doi/full/10.1126/science.1214025>

Pérez et al (2021), <https://pubs.acs.org/doi/10.1021/jacs.1c04152>

Referee #2 (Remarks to the Author):

My expertise is in molecular evolution, I will thus submit my evaluation of the phylogenetics based evolutionary claims declared in the manuscript. Overall, the general evolutionary and phylogeny framing of the paper (and I believe one of the main reasons behind the claimed novelty) has several flaws.

To start, the connection between methanogenesis biosynthesis proteins and nitrogenase is not new (e.g., see Boyd et al), and it's also known that *nifB* is needed to support methanogenesis. The claim that the earliest role of *nifB* was for methanogenesis (instead of N-fixation) is new, however there are major issues with their tree reconstructions and inferences thus this new claim lacks supporting evidence. At a top level, unfortunately, the tree reconstruction seems sloppy and needs more care; a well-rooted *nifB* tree could provide novel evidence here, but I can't understand their description of the rooting method and their phylogeny-based claims seems possibly circular. Based on the figures (they are a little unclear and hard to read, I did my best) methanogens are placed in the deepest branching lineages, so naturally one would infer an early methanogen-associated role for *nifB*.

Similarly, the rest of this section relating to *CfbD* is also not new (see again Boyd et al. *Geobio* 2011), and there isn't any phylogenetic evidence presented in the paper specifically to support the rooting for that either. In general, they would have to make some other independent and convincing rationale for the rooting. The conclusions are bold and flashy but naturally one looks at the data (i.e., the tree) as they currently stand aren't the best evidence.

We are very thankful with the reviewer, and we have taken these concerns seriously to present a substantially more strengthened argument. We now present phylogenies of McrC and Mmp7, which bind the potential [8Fe-9S-C] clusters (or L-clusters) in our structures (Extended Data Fig. 9a,b). These proteins have topologies very similar to other published MCR phylogenies and, by comparison with the archaeal species phylogeny, allowed us to place both McrC and Mmp7 proteins in the last common ancestor of TACK and Euryarchaeota. This is consistent with multiple recent studies which place MCR's catalytic subunits in this ancestor as well (Hua et al 2019; Garcia et al 2022; Mei et al 2023).

We also inferred an expanded NifB phylogeny and found this protein in TACK, Asgard and Euryarchaeota (Extended Data Fig. 9c). Together with previously proposed evidence of NifB as necessary for methanogenesis (Fay et al 2015; Lei et al 2022; Saini et al 2023) – with our work implying that this is because of the L-clusters required to activate MCR – NifB is placed at the last common ancestor of TACK and Euryarchaeota. The latter now leads to question whether L- or M-cluster bearing members of the nitrogenase family also existed this early. We approached this problem in two steps. First, we inferred a phylogeny of CfbD – a known distant homolog of nitrogenases crucial in the biosynthesis of MCR's catalytic coenzyme F₄₃₀ (Extended Data Fig. 9d). Unsurprisingly, CfbD (which does not bind L-clusters) is also found in TACK, Asgard-, and Euryarchaeota, making it as ancient as MCR and its activation machinery. For the second step, we inferred a separate phylogeny of the wider nitrogenase family (Extended Data Fig. 10b), using all members that are either known or suspected to interact with L-clusters such as: classical nitrogenases and their maturases, group IV nitrogenases recently shown to contain an L-cluster in their active sites (i.e., methylthio-alkane reductases, Lago-Maciel et al. 2024), group IV and VI nitrogenases with unknown function (i.e., NfID, North et al 2020), and specialized maturases for the vanadium nitrogenase. All these proteins share a particular architecture that is different from CfbD: they are heterotetramers of two paralogs (usually designated as D and K). This difference in oligomerization is the result of a series of gene and operon duplications, which have been extensively described in several publications (Fani et al 2000; Raymond et al 2004; Boyd et al 2011). CfbD, on the other hand, has not undergone such duplications and only forms a homodimer (Extended Data Fig. 10a). For this reason, CfbD is the outgroup (using duplication rooting) for the entire Nif family. Thus, our argument was to root this phylogeny between all D and K subunits, using in effect the very first gene duplication in this family to root the tree.

In our phylogenetic tree of the Nif family, we only found very sporadic evidence of TACK or Asgard archaea despite extensive blast searches (Extended Data Fig. 10b). In each case, these sequences nest within groups of Euryarchaeota or Bacteria, which implies that the few TACK and Asgard archaea that do have these proteins received them horizontally. In contrast, Euryarchaeota are well represented in paralogs of nitrogenases. We interpret these data to mean that heterotetrameric members of the Nif family (known or suspected to interact with L- and M-clusters) evolved somewhere in the Euryarchaeota, after this group had split from TACK and Asgard archaea. This is consistent with prior publications, which the reviewer also drew our attention to (i.e., Boyd et al 2011). Therefore, we can place L-cluster binding proteins of the MCR activation machinery at an earlier node of the Archaeal species phylogeny in contrast to proteins of the Nif family (except for CfbD, which does exist in TACK and Asgard archaea but does not bind L-clusters).

Further details:

“A different root would imply more horizontal gene transfers between archaea and, consequently, becomes less likely”
What is the evidence for this?

“After mapping the presence of the nitrogenase catalytic subunits NifK and NifD29-31 onto our tree, the data suggests that nitrogenase was absent at the root node of the NifB phylogeny. This supports the idea that NifB’s role during MCR activation precedes its association with nitrogenases.”

It’s not obvious to me looking at Extended Data Fig. 15 that nitrogenases were absent in organisms with oldest ancestral NifB, rather than secondarily lost in some archaeal lineages. Further, it’s unclear from the methods whether amplicons/incomplete genome assemblies were included as sources of NifB sequences. Therefore, the missing nitrogenase genes is likely an artifact of the sequence search strategy. Lastly, make sure your represented taxa is available and easy to access, it is extremely hard to replicate/understand their analyses without this.

We appreciate the reviewer for pointing at these issues, some of which we agree required more details to improve readability and understanding of our ideas. As a clarification, we utilized refseq for our blast queries, which includes incomplete genomes. While it is not that incompleteness is the reason we can easily find NifB and CfbD in TACK and Asgard archaea, but other members of the Nif family only sporadically, we find this very implausible. We therefore do not agree that missing nitrogenases are a result of our search strategy.

We unfortunately do not share the reviewer’s observation that missing nitrogenases in TACK and Asgard archaea imply losses. As it stands, our results provide evidence of MCR being older than all L- or M-cluster bearing members of the nitrogenases. For MCR and such relevant Nif proteins to be of equal age requires additional gene losses as the reviewer points out. We do, however, fully acknowledge that discoveries of new genomes and sequences could in principle overturn our conclusions, as is always the case in deep evolutionary history. We have therefore adjusted our wording in the text to indicate that current evidence favours the first use of L-clusters to have been in MCR and methanogenesis, not in the nitrogenase family.

To improve accessibility to our data, we have additionally uploaded all necessary sequence identifiers employed for our phylogenetic trees along with this submission.

“Inferring from our NifB phylogeny that methanogenesis originated earlier than nitrogen fixation...” Because the rooting method for the NifB tree is unclear (see first comment), I’m not convinced that this conclusion is specifically supported by the NifB tree.

We now base this statement on a much wider set of observations, as detailed in our previous comments.

“...we then rooted this tree with CfbD. These results imply that nitrogenases have evolved from enzymes able to convert porphyrin-based substrates” It has previously been established that CfbD is homologous to NifDKEN, and it’s also previously been proposed that a CfbD-like protein was ancestral to NifDKEN (e.g., see Boyd et al., *Geobiology*, 2011). It’s therefore misleading to claim that this is a new finding. In addition, from the preceding comment, it is not appropriate to root the nitrogenase tree with CfbD. An independent rationale for the rooting is needed to support their flashy claims.

As we have described in our explanation above, we consider CfbD as the only appropriate root of the overall nitrogenase family. As indicated by the reviewer, CfbD is a known ancestral homologous, but structurally dissimilar to other members of the nitrogenase family. While CfbD only oligomerize as a homodimer, all other known members of the nitrogenase family form heterotetramers of two paralogs (normally D and K). These paralogs evolved from a series of duplications. An initial gene duplication created an original pair of paralogs, which presumably could have already formed a heterotetramer. This original pair then underwent multiple operon duplications, resulting in the wide diversity of nitrogenase family members we know today (bona fide nitrogenases and their maturases, group IV and group VI nitrogenases, etc.). When we root within CfbD, one original duplication is required in a member of Euryarchaeota, followed by the subsequent operon duplications that explain the trees. Rooting within other members of the wider Nif family (as opposed to CfbD) requires one original duplication, then a loss of a missing CfbD paralog, as well as

independent losses of Nif proteins in Asgard and TACK archaea, plus all the operon duplications enlisted before. A root in CfbD is, therefore, a much more parsimonious hypothesis.

To summarize: We believe the root lies in CfbD for two major reasons. First, because CfbD is present within a wider taxonomic range of archaea (TACK, Asgard, and Euryarchaea, which branch broadly in accordance with the species phylogeny) in comparison to other Nif proteins. And second, rooting in CfbD requires fewer gene duplications and losses than rooting within the other paralogs in the wider Nif phylogeny. It thereby represents the most parsimonious root with respect to the implied number of gene duplications and losses.

“Together, our results strongly suggest that the M-clusters of nitrogenases originate from the methanogenic FeS clusters in the MCR activation complex.” Metalloclusters don’t have “ancestry”, the genes that make them do.

Point taken. We apologize for the incorrect use of words as we meant that the gene that synthesizes these clusters first evolved in the context of methanogenesis, and that the L-cluster first function was to participate in MCR’s activation complex. This has now been corrected.

References:

- Hua et al 2019, <https://www.nature.com/articles/s41467-019-12574-y>
Garcia et al 2022, <https://www.annualreviews.org/content/journals/10.1146/annurev-micro-041020-024935>
Mei et al 2023, <https://academic.oup.com/pnasnexus/article/2/2/pgad023/7010768>
Fay et al 2015, <https://www.pnas.org/doi/10.1073/pnas.1510409112>
Lei et al 2022, <https://journals.asm.org/doi/full/10.1128/spectrum.02093-21>
Saini et al 2023, <https://www.biorxiv.org/content/10.1101/2023.10.20.563283v1>
Fani et al 2000, <https://link.springer.com/article/10.1007/s002390010061>
Raymond et al 2004, <https://academic.oup.com/mbe/article/21/3/541/1079575?login=false>
Lago-Maciel et al 2024, <https://www.biorxiv.org/content/10.1101/2024.10.19.619033v1>
North et al 2020, <https://www.science.org/doi/10.1126/science.abb6310>
Boyd et al 2011, <https://onlinelibrary.wiley.com/doi/full/10.1111/j.1472-4669.2011.00278.x>

Referee #3 (Remarks to the Author):

The manuscript by Ramirez-Amador et al of the ATP-dependent methyl-coenzyme M reductase activation complex reveals the structure of the complex responsible for the reductive activation of F₄₃₀ to the Ni(I) state. Three FeS clusters are present in the activation complex that astonishingly resemble the nitrogenase FeMo-cofactor; these clusters form an electron transfer pathway towards F₄₃₀. This is the first example of this type of metallocluster outside of nitrogenase and is an extraordinary observation with potentially far-reaching implications for the evolution of the biosynthetic pathway for this cluster. Present in the activation complex is the ATP-binding cassette (ABC) fold containing A2 subunit that likely is involved in rearrangements powered by ATP binding and hydrolysis. Overall, this is remarkable study and I enthusiastically support publication.

A few comments for the authors consideration:

line 112 - how does this activity (16 nmol CH₄/120 min/10 ug) compare to that reported for the specific activity of MCR?

We appreciate the reviewer's comments. Since this is the first report on the activity of MCR coming from the mesophile Methanococcus maripaludis, there is no other value in literature for direct comparison. Additionally, we show the activity values of a transitory low-abundance intermediate of MCR bound to the activation complex. However, we now added a statement where we mention the activity values of MCR from Methanothermobacter marburgensis and Methanothermobacter thermoautotrophicus. Here, we also shortly detail the small but key differences with our experimental set up.

Addition in the main text:

"Upon the addition of Ti(III)-citrate, earlier studies on the activation of MCR achieved a maximum activity of 0.1 μmol CH₄ min⁻¹ mg⁻¹ (or 0.1 U/mg), not much different from our results after 2 hours of reaction. Nevertheless, such values remain low when compared to the average activity of mature MCR from Methanothermobacter species (20 U/mg, being 100 U/mg the maximum ever reported). On the other hand, our purification strategy proved to be successful capturing a low-abundance and highly transient biogenesis intermediate, challenging a direct comparison with a fully mature MCR. Such discrepancy also indicates an incomplete understanding of the activation process, insinuating that additional factors might still be required to efficiently reduce the nickel ion in F₄₃₀."

line 131 - "one of the MCR heterodimers" - should this be heterotrimer?

Corrected. Thank you for the observation.

line 143 - another example of an enzyme with an ABC subunit is the carbon-phosphorus lyase reported by Amstrup et al Nat. Comm. 14, 1001 (2023). In this system, "ATP hydrolysis induces drastic structural remodeling leading to opening of the core complex and reconfiguration of a metal-binding and putative active site", not unlikely what is proposed here.

line 282 - the structure of the ADP-AIF stabilized nitrogenase complex provided the first demonstration of how nitrogenase operates as a nucleotide dependent switch (Schindelin et al Nature 387, 370 (1997)). While both nitrogenase and ABC proteins couple nucleotide binding and hydrolysis to protein conformational changes, they have distinct protein folds. The archerases discovered by W. Buckel (see FEMS Microbiol. Rev. 28, 455 (2004)) should also be included in this list.

We are thankful to the reviewer for bringing up these important examples. As part of our discussion section, we have now included both PhnK-CPL complex and archerases as examples of other ABC modules and ATP-switches, respectively. Both thoroughly depict our idea of ATP-induced responses that directly impact enzymatic catalysis of large protein complexes. Additionally, we have also included the corrinoid cofactor of O-demethylases in this list to strengthen our argument on the ATP-mediated modulation of the redox potential in porphyrin coenzymes.

Addition in the main text:

“However, the binding and hydrolysis of ATP could theoretically increase the redox potential in a way that Ti(III)-citrate ($E^{\circ} = -480$ mV at pH 7) is able to reduce Ni(II) to Ni(I). In a similar fashion, during the ATP-dependent activation of some corrinoid cofactors, ATP hydrolysis drastically increases the midpoint redox potential of the Co(II)/Co(I) couple (-500 to -250 mV). Moreover, additional Mmp interactors may push the redox potential of F_{430} even further, potentially allowing the reduction of nickel by milder reducing agents like DTT ($E^{\circ} = -320$ mV) and physiologically relevant electron donors such as reduced ferredoxin ($E^{\circ} = -420$ mV). Altogether, this places the A2 component’s function on the activation of MCR alongside other prominent nucleotide-switching machines able to drive catalytic activation or product release.”

line 188 - as near as I can tell from Figure 4 and extended Figure 13, the densities for FeSI, II and III clearly resemble that for the nitrogenase FeMo-cofactor. However, without looking at the actual maps, it is not possible to make this assessment with full confidence.

Among our supplementary files we have supplied all the cryoEM electron density maps that were utilized for this study. Particularly, sharpened maps allow to observe the density of all ligands in great detail – including all three FeS clusters. This, in combination with our improvements to reach higher local resolutions (1.8 to 2.1 Å), makes us confident on the assignment of the L-cluster topology shown by all metallocofactors in the activation complex.

line 250 - has the consequence of a NifB deletion on the MCR activation machinery been studied? Since the native complex is tagged with Twin-Strep tag using the CRISPR-Cas12a system, this should be possible.

*The effect of a $\Delta nifB$ strain on *M. maripaludis* has not been investigated. However, on this regard, evidence provided by Saini et al (2023) has shown the plasticity of *Methanosarcina acetivorans* to avoid full inactivation of NifB. Here, the authors demonstrate that a complete abolishment of *nifB* is not possible unless plasmid complementation carrying a heterologous *nifB* takes place to produce the L-clusters (*M. acetivorans* is a non-diazotrophic methanogen). This way, an essential but yet unknown role of NifB beyond nitrogen fixation was proposed. Similar results have been observed when trying to delete essential genes in strains of *Methanococcus maripaludis* (Stock et al 2010). Beyond such mutation buffering capacity, we believe that studies to characterize the role of NifB in the activation of methanogenesis may elaborate a more detailed follow-up story dedicated mostly to such physiological implications.*

As a general comment for speculation - why are 3 copies of such an elaborate metallocluster be needed to activate F_{430} - are they "only" serving as electron carriers or might they have more complicated roles?

Given the architecture of our structure, FeSIII is the cofactor that is most surface exposed, potentially serving as initial electron acceptor. We believe that other redox proteins may participate during the activation of F_{430} (i.e., polyferredoxin or heterodisulfide reductase) with FeSIII. Thus, multiple electron transfer modules may shape a direct route from the electron donor towards F_{430} across all three newly discovered L-cluster-like cofactors. On the other hand, although L-clusters share similar electronic features to FeMoco and have shown reduced catalytic activity in NifEN, the coordinating residues within McrC and Mmp7 differ from those known in nitrogenase. We believe that these differences in amino acids are mostly for modulation of the redox potential and these FeS clusters might be limited to function as electron carriers. Nonetheless, deeper studies are required to rule out other potential roles in the physiology of methanogens. (Jasniewski et al 2019; Hu et al 2009; Lee et al 2024).

Fig. 4b. FeS clusters in their electron-density with their coordinating residues (contour level σ 0.168 for FeSI and σ 0.235 for FeSII and FeSIII).

References:

- Saini et al (2023), <https://www.biorxiv.org/content/10.1101/2023.10.20.563283v1>
 Stock et al (2010), <https://onlinelibrary.wiley.com/doi/10.1111/j.1365-2958.2009.06970.x>
 Jasniewski et al (2019), <https://onlinelibrary.wiley.com/doi/10.1002/anie.201907593>
 Hu et al (2009), <https://www.pnas.org/doi/10.1073/pnas.0907872106>
 Lee et al (2024), <https://www.science.org/doi/epdf/10.1126/sciadv.ado6169>

Referee #1 (Remarks to the Author):

In the previous version of the manuscript by Ramirez-Amador et al. there were concerns raised regarding the ATP-dependent activity assays of the activation complex, specifically regarding the low specific activity of the activated MCR, the small observed rate stimulation by ATP, the presence of MCRox1 in the sample (which can be activated by the reductant directly), and the lack of a control in the absence of the activation complex. These concerns were not completely addressed. However, the addition of a control where alkaline phosphatase was included in the reaction mixture (as well as mass photometry experiments) provides strong support that the reductive activation was indeed ATP dependent and the activity derived from the direct activation of MCRox1 was negligible.

Concerns were also raised regarding the definitive assignment of the structures of the complex iron sulfur clusters, i.e., whether there was a central carbide similar to the structure of FeMoco in nitrogenase. While the suggested, more definitive methods of confirmation were not pursued, additional local refinement of the structure and the inclusion of a figure showing the density and distances of the atoms in the clusters and a table comparing these with FeMoco, FeFeco, and the L- and K-clusters help to make the argument more compelling (Extended Data Table 5). Together these additions have improved the manuscript such that it is suitable for publication in Nature. Below are some minor items to address prior to publication.

It is stated in the Methods section and rebuttal that resolutions up to 1.8 Å were obtained in the newly refined data but the in the Abstract it still says 2.1 Å.

We truly appreciate the referee's comments. We have now described a resolution interval (1.8 to 2.1 Å) in the abstract.

Line 52: The structure of the macrocycle of the F430 is technically not a porphyrin.

We refer now to F430 as a porphyrin-based molecule in this section.

Line 55: F430 can exist in +1, +2, and +3 oxidation states but does not transition between all three during the catalytic cycle.

We have now updated this sentence as it follows:

"Here, the nickel ion of F430 can exist in the oxidation states +1, +2, and +3 but, to initiate its catalytic cycle, nickel must be in the Ni(I) state (MCRred1)..."

Line 80: Activation relies on ATP hydrolysis (not relays).

Corrected.

Line 110: The Ni(III)-MCRox1 state not only is not ruled out by the UV-vis data, its presence in the activation complex sample was confirmed via EPR (Fig. 4d). Thus stating that it is below the detection limit is misleading.

Corrected. The sentence has been slightly modified as it follows:

"However, Ni(III)-MCRox1 features in the UV-vis spectrum cannot be discarded either (see below)..."

Here, '(see below)' stands for the EPR data described later in the text.

Line 133: Similar to the catalytic cycle of nitrogenase, ATP binding, hydrolysis, and release is likely coupled to MCR activation complex formation, F430 reduction, and complex dissociation. The mass photometry

data does suggest that ATP binding promotes complex formation, but it is somewhat confusing to say it doesn't play a role in the dissociation of the complex (and by itself does not rule out that the direct reduction of MCRox1 could be occurring).

Corrected. To avoid misleading interpretations the sentence has been modified as it follows:

"This demonstrates that ATP-binding promotes the assembly of the A2 component with MCR, reinforcing A2's crucial function for full complex formation. ..."

Line 221: May be less confusing to refer to the clusters as [8Fe-9S-C] as opposed to double cubane [4Fe-4S] clusters.

Corrected. The line has been slightly edited as it follows:

"An average of 27 ± 2 Fe ions, identified by ICP-MS (Supplementary Table 4), supports the presence of three [8Fe-9S-C] clusters..."

Line 310: You should include the actual calculated specific activity of your enzyme preparations for comparison. If my calculations were correct, the highest activation that was observed was approximately 5-fold less than that from the A3a complex.

We have now added the specific activity value (0.028 U/mg) in the discussions section.

Fig. 4. Panel d mislabeled as f.

Corrected.

Referee #2 (Remarks to the Author):

I don't have any further comments.

We thank the referee.

Referee #3 (Remarks to the Author):

In this revised manuscript, the authors have satisfactorily addressed my concerns, and I continue to be enthusiastic about this work.

We thank the referee.